



# Lake-TopoCat: A global lake drainage topology and catchment database

Md Safat Sikder[1], Jida Wang[1], George H. Allen[2], Yongwei Sheng[3], Dai Yamazaki[4], Chunqiao Song[5], Meng Ding[1], Jean-François Crétaux[6], and Tamlin M. Pavelsky[7]

[1]Department of Geography and Geospatial Sciences, Kansas State University, Manhattan, KS, USA
[2]Department of Geosciences, Virginia Tech, Blacksburg, VA, USA
[3]Department of Geography, University of California, Los Angeles, CA, USA
[4]Institute of Industrial Science, The University of Tokyo, Tokyo, Japan
[5]Nanjing Institute of Geography and Limnology, Chinese Academy of Sciences, Nanjing, China
[6]Laboratoire d'Études en Géophysique et Océanographie Spatiales (LEGOS), Centre National d'Études Spatiales (CNES), Toulouse, France
[7]Department of Earth, Marine and Environmental Sciences, University of North Carolina, Chapel Hill, NC, USA

*Correspondence to*: Md Safat Sikder (msikder@ksu.edu); Jida Wang (jidawang@ksu.edu)

**Abstract.** Lakes and reservoirs are ubiquitous across global landscapes, functioning as the largest repository of liquid surface
freshwater, hotspots of carbon cycling, and "sentinels" of climate change. Although typically considered as lentic (hydrologically stationary) environments, lakes are an integral part of global drainage networks. Through perennial and intermittent hydrological connections, lakes often communicate with each other, and these connections actively affect water mass, quality, and energy balances in both lacustrine and fluvial systems. Deciphering how global lakes are hydrologically interconnected, or the so-called "lake drainage topology", is not only important to lake change attribution, but also increasingly
critical to discharge, sediment, and carbon modeling. Despite the proliferation of river hydrography data, lakes remain poorly represented in routing models, partially because there has been no global-scale hydrography dataset tailored to lake drainage basins and networks. Here, we introduce the global Lake drainage Topology and Catchment database, or "Lake-TopoCat", which reveals detailed lake hydrography information with a careful consideration of possible multifurcation. Lake-TopoCat contains the outlet(s) and catchment(s) of each lake, the inter-connecting reaches among lakes, and a wide suite of attributes
depicting lake drainage topology such as upstream and downstream relationship, drainage distance between lakes, and a priori drainage type and connectivity with river networks. Using the HydroLAKES (v1.0) global lake mask, the Lake-TopoCat v1.0 identifies ~1.46 million outlets for ~1.43 million lakes larger than 10 ha and delineates 77.5 million $km^2$ of lake catchments covering 57% of the Earth's landmass except Antarctica. The global lakes are interconnected by ~3 million reaches, derived from MERIT Hydro (v1.0.1), stretching a total distance of ~10 million km, ~80% of which are shorter than 10 km. With such
unprecedented lake hydrography details, Lake-TopoCat may facilitate a variety of limnological applications including water quality diagnosis, agriculture and fisheries, lacustrine connectivity monitoring, and integrated lake-river modeling. It is freely accessible at https://doi.org/10.5281/zenodo.7420810 (Sikder et al., 2022).



# 1 Introduction

Natural lakes ponds, and human-made reservoirs, hereafter "lakes", store the largest amount of liquid freshwater on Earth's surface (Oki and Kanae, 2006; Abbott et al., 2019). Although widely perceived as water stores and lentic systems, lakes are often connected to each other through river networks and are inherent components of the global drainage system (Fergus et al., 2017; Gardner et al., 2019). The water balance in a lake and its water quality (e.g., turbidity, dissolved oxygen, acidity, and temperature) reflect climatic, hydrological, geologic, and land use characteristics of the local drainage catchment (Yang et al., 2022). The limnological properties of one lake also affects others via the transfers of water mass (e.g., Huziy and Sushama, 2017), sediments, nutrients (e.g., Stieglitz et al, 2003), and energy along connecting rivers. Since lakes sequester a large amount of organic and inorganic carbon (Tranvik et al., 2009; Mendonça et al., 2017), deciphering how global lakes are connected through drainage networks will complement routing models, which currently emphasize rivers (Leibowitz et al., 2018; Liu et al., 2022), to better constrain the terrestrial carbon cycling (Cardille et al., 2007). The drainage paths between lakes, including their distances, gradients, connectivity, and climatic zonation, also determine the migration pattern for lacustrine species, which is crucial for monitoring ecosystem health and services under climate change (Woolway et al., 2020). For these reasons, a hydrography dataset tailored to global lakes, which provides fine spatial details of the boundary of each lake catchment and how lakes are topologically interconnected, is overdue and has the potential for high hydrological and ecological significance.

In recent years, significant improvements have been achieved in global lake mapping and inventories. A few prominent examples are HydroLAKES (v1.0) which inventories 1.4 million lakes larger than 10 ha (Messager et al., 2016), the GLAKES comprising 3.4 million lakes with maximum surface area larger than 3 ha (Pi et al., 2022), and the UCLA circa-2000 and circa-2015 Global Lake Inventories with more than 9 million quality-assured lake polygons larger than 0.4 ha (Sheng et al., 2016). These water masks, unfortunately, emphasize individual lake entities and provide little to no metadata depicting drainage hydrography (such as catchment boundaries and hydrological connectivity) among the lakes. The lack of drainage hydrography has restricted application of these refined lake inventories to watershed-scale processes and has left lake-river integration a challenging task.

Only a few attempts have been made to integrate global lakes to drainage networks. In the "customized" format of HydroBASINS (Lehner and Grill, 2013), for example, lake polygons were directly clipped into the river sub-basin polygons, in order to register the lakes to the topological (upstream and downstream) coding structure of HydroBASINS. Nevertheless, since the original sub-basins in HydroBASINS were generated without consideration of lake presence, there was no guarantee that the sub-basin pour points are aligned with the lake inlets or outlets. This inconsistency leads to cases where a sub-basin polygon contains multiple lakes or a lake polygon contains the pour point of the most downstream intersecting sub-basin. In such cases, the drainage topology among lakes is incompletely depicted, and the intersecting subbasins may underrepresent or overshoot the spatial domains of the lake catchments. Similar issues were also acknowledged in the Global Lake area, Climate,





and Population dataset (GLCP), where HydroBASINS was also used to match lakes to the surrounding sub-basins (Meyer et al., 2020).

In HydroLAKES (Messager et al., 2016) and the most recent LakeATLAS dataset (Lehner et al., 2022), the outlet and the upstream drainage area associated with each lake were derived from the 15 arc-second HydroSHEDS hydrography dataset (Lehner et al., 2008). Their methods of lake outlet delineation, however, excluded the possibility of lake bifurcation or

multifurcation (i.e., a lake flowing out in two or more directions), and the 15 arc-second resolution (about 450 m at the equator) of HydroSHEDS might be too coarse to derive reliable catchment boundaries for many small lakes in HydroLAKES (with a minimum size of 0.1 km$^2$). In addition, lake catchments in HydroLAKES or LakeATLAS were provided with only area values, not geometric boundaries, meaning that spatially explicit applications of such catchment information remain inconvenient. In fact, despite the proliferation of river basin datasets (e.g., Lehner and Grill, 2013; Lin et al., 2021), there has been no global-

scale catchment data tailored specifically for lakes. To our knowledge, only a few regional lake watershed datasets are available so far, and examples are the National Hydrography Dataset Plus Version 2 (NHDPlusV2) (McKay et al., 2012) and the Lake-Catchment (LakeCat) Dataset (Hill et al., 2018) for the US, the COmprehensive Data set for China's Lake Basins (CODCLAB) for 767 large lakes (>10 km$^2$) in China (Chen et al., 2022a), and several lake catchment datasets for the endorheic Tibetan Plateau (e.g., Liu et al., 2020 and 2021; Yan et al., 2019).

More detailed configurations of lake drainage connectivity are also limited to regional scales. For the US, Schmadel et al. (2018) and Gardner et al. (2019) integrated lakes into river networks as depicted in the NHD. In these studies, lakes were considered as part of the drainage system only if the lake polygon was directly intersected by river channels. While this is reasonable in many cases, hydrological connectivity among lakes can be established through intermittent channels, ephemeral flow (such as via "fill and spill"), and subsurface flow (Leibowitz et al., 2016; McDonnell et al., 2021), which are not always

visible or channelized in the NHD vector networks. Alternatively, multitemporal spectral images have been applied to better capture the variability in intermittent connectivity. For instance, Vanderhoof et al. (2016 and 2017) used Landsat images to examine the patterns of wetland connections in relation to wetland arrangements and surface water expansion across the Prairie Pothole Region of the US. Tan et al. (2019) used Landsat and MODIS imagery to quantify the surface water connectivity to understand the complex surface water dynamics of Poyang Lake in China. More recently, Dolan et al. (2021) applied Landsat-

observed sediments in water to understand the variation in lake-to-channel connectivity and the impact on ice phenology within the Colville River Delta, Alaska. The completeness and consistency of the observed lake-river connectivity are, however, affected by the image quality and resolution. This is particularly problematic when channel visibility is obstructed by riparian canopies. In addition, although higher-resolution images may improve the mapping of lake-river connections (e.g., Wu and Lane, 2017), optical images alone are not yet sufficient to decipher complex drainage topology among global lakes.

We argue that, while lake-river connectivity may vary through time, a prerequisite for monitoring such important dynamics is to sort out the drainage topology (i.e., upstream and downstream relationships) among global lakes based on their potential connecting paths. This lake topology, albeit temporarily static, ensures all lakes on the continental surface to be registered to the drainage system, and will serve as useful a priori networks for examining the temporal variability in surface water

Earth System
Science
Data

connection. This topology will also facilitate a more thorough inclusion of lakes into global hydrological models. For example,

Bowling and Lettenmaier (2010) showed that lakes in the Arctic region can store up to 80% of the snowmelt water each year, ultimately reducing the spring peak of river flows in that region. Considering the unique roles of lakes, numerous studies have incorporated lakes into hydrological and/or hydrodynamic models to improve the accuracy of simulated streamflow (e.g., Han et al., 2020; Lin et al., 2015), water temperature (Tokuda et al., 2021), and regional climate (Huziy and Sushama, 2017). Lakes can be registered manually into the routing networks in case of small areas, where the number of lakes is manageable (e.g.,

Lin et al., 2015). On the other hand, usually only large lakes have been resolved in continental- or global-scale modeling networks (e.g., Tokuda et al., 2021), due to the constraints of model and hydrography data resolutions, computing power, and additional workload.

To address these gaps, we introduce the global Lake drainage Topology and Catchment database or Lake-TopoCat. Lake-TopoCat v1.0 was constructed for nearly all 1.4 million lakes in HydroLAKES v1.0, but our automated algorithm is generic

and applicable to other global lake inventories (see Section "Database availability"). Lake-TopoCat offers an unprecedented detail of global lake hydrography, including spatially resolved perimeters of the lake catchments, drainage outlets from the lakes, perennial, intermittent, and potential drainage reaches between the lakes, and importantly, the topological relationships among the lake, catchment, and reach structures. To tackle the above-described challenges, we applied the MERIT Hydro high-resolution raster hydrography maps (Yamazaki et al., 2019). Our algorithm also included a mechanism to capture possible

lake bifurcation. The remaining sections will describe the data structure of Lake-TopoCat, followed by input and validation data, methodology, quality assessment, and technical validation. We conclude with a discussion of Lake-TopoCat's potential applications and limitations.

## 2 Data description and structure

Lake-TopoCat v1.0 (hereafter referred as TopoCat) consists of five feature components, each with multiple attributes depicting

lake drainage relationships. The five features are (1) lake boundaries (polygons), (2) lake outlets (points), (3) unit catchment boundaries (polygons) defining the drainage areas between cascading (i.e., immediately upstream and downstream, see definition in Section 2.2 and 2.3) lake outlets, (4) inter-lake reaches (lines) defining the drainage networks that connect the lake outlets to the inland sinks or the ocean, and (5) lake-network basins (polygons) that define the entirety of the drainage area containing each inter-lake network (i.e., a complete basin from the headwater to an inland sink or the ocean for all basins

containing lakes). An example of the feature components is given for a focal region (about 6300 km$^2$) in the Canadian Shield (Fig. 1). The attribute tables for each of the feature components are explained in Table 1. Each of the components and their associated attributes are described in further detail below.




**Table 1. Attribute definitions in each of the five feature components of TopoCat.** Here, dd, and masl are abbreviations of decimal degrees, and meters above sea level, respectively.

| Feature | Attribute | Description | Unit |
|---|---|---|---|
| **Lakes**: Individual lake polygons as in HydroLAKES. File name: Lakes_pfaf_xx, where "pfaf_xx" indicates the IDs of the 68 Pfafstetter level-2 basins for data organization (see Section 4). | Hylak_id | HydroLAKES ID for this lake | N/A |
| | Lake_area | Lake water surface area | km$^2$ |
| | Outlet_n | Count of outlets of this lake (>1 indicating multifurcation) | N/A |
| | D_hylak_id | List of IDs of the next (i.e., directly connected) downstream lakes | N/A |
| | D_lake_n | Count of the next downstream lakes | N/A |
| | D_lak_ntot | Count of all downstream lakes to the sink or ocean, excluding this lake | N/A |
| | U_lake_n | Count of the next upstream lakes | N/A |
| | U_lak_ntot | Count of all upstream lakes from the headwater, excluding this lake | N/A |
| | Cat_a_lake | Area of the total upstream catchment(s) (accumulative from the headwater) for this lake. In case of multifurcation, this value aggregates the areas of the upstream catchments for all outlets of this lake. | km$^2$ |
| | Lake_type | Drainage type of this lake in relation to the inter-lake reach network: headwater, flow-through, terminal, or coastal. | N/A |
| | Lake_order | Strahler order of the lake in the inter-lake reach network | N/A |
| | Laktyp_mhv | Drainage type of this lake in relation to rivers in MERIT Hydro-Vector: isolated, headwater, inflow-headwater, flow-through, terminal, or coastal. | N/A |
| | Basin_id | ID of the lake-network basin (see below) this lake belongs to | N/A |
| **Outlets**: Outlet or pour points from each of the lakes. There are multiple outlets from a multifurcation lake. File name: Outlets_pfaf_xx | Outlet_id | ID of this lake outlet, which is different from Hylak_ID as there can be more than one outlet for a multifurcation lake. | N/A |
| | Hylak_id | ID of the associated HydroLAKES lake | N/A |
| | Outlet_lat | Latitude at this outlet | dd |
| | Outlet_lon | Longitude at this outlet | dd |
| | Outlet_elv | Elevation at this outlet (based on hydrologically adjusted MERIT DEM, as for other elevation values) | masl |
| | D_out_id | ID of the next downstream outlet | N/A |
| | D_hylak_id | ID of the next downstream lake | N/A |
| | D_lak_ntot | Count of all downstream lakes to the sink or ocean, excluding this lake | N/A |
| | D_reach_id | ID of the connecting downstream reach | N/A |
| | D_slope | Hydraulic gradient (slope) from this outlet to the next downstream outlet or sink | N/A |
| | D_distance | Drainage distance from this outlet to the next downstream outlet or sink | m |
| | D_dst_sink | Total drainage distance to the sink or ocean | m |
| | U_lake_n | Count of the next upstream lakes | N/A |
| | U_dist_avg | Mean distance from the next upstream outlets | m |
| | U_dist_min | Minimum distance from the next upstream outlets | m |
| | U_dist_max | Maximum distance from the next upstream outlets | m |
| | U_lak_ntot | Count of all upstream lakes from the headwater, excluding this lake | N/A |
| | Cat_area | Area of the associated unit catchment | km$^2$ |





| | Cat_a_tot | Area of the entire upstream drainage basin, i.e., from the headwater to this outlet | km² |
|---|---|---|---|
| | Out_type | Drainage type of this lake outlet in relation to the inter-lake reach network: headwater, flow-through, terminal, or coastal. | N/A |
| | Out_order | Strahler order of the outlet in the inter-lake reach network | N/A |
| | Basin_id | ID of the lake-network basin this outlet belongs to | N/A |
| **Unit catchments**: Boundaries of the drainage area between each lake outlet and the next upstream outlet(s), or the drainage area from the headwater to the outlet if there are no lakes upstream. File name: Catchments_pfaf_xx | Outlet_id | ID of the associated outlet, also used as the ID of the unit catchment. | N/A |
| | Hylak_id | ID of the associated lake | N/A |
| | D_out_id | ID of the next downstream outlet | N/A |
| | D_hylak_id | ID of the next downstream lake | N/A |
| | Cat_area | Area of this unit catchment | km² |
| | Cat_type | Drainage type of this unit catchment in relation to the inter-lake reach network (same as in "Outlet_type"). | N/A |
| | Basin_id | ID of the lake-network basin this unit catchment belongs to | N/A |
| **Inter-lake reaches**: Drainage networks inter-connecting the lake outlets to the next downstream lake outlets, inland sinks, or the ocean. File name: Reaches_pfaf_xx | Reach_id | ID of this inter-lake reach | N/A |
| | D_reach_id | ID of the next downstream reach | N/A |
| | D_out_id | ID of the connected downstream outlet | N/A |
| | D_hylak_id | ID of the connected downstream lake | N/A |
| | U_out_id | ID of the connected upstream outlet | N/A |
| | U_hylak_id | ID of the connected upstream lake | N/A |
| | U_lak_ntot | Count of the total upstream lakes from the headwater | N/A |
| | Start_elv | Elevation at the starting node of this reach | masl |
| | End_elv | Elevation at the ending node of this reach | masl |
| | Rch_length | Reach geodesic length | m |
| | Rch_slope | Hydraulic gradient (slope) of this reach | N/A |
| | Rchint_mhv | Proportion of this inter-lake reach (in % length) overlapped by river channels in MERIT Hydro-Vector, implying the likelihood of this reach being more perennial or ephemeral | N/A |
| | Rch_order | Strahler order of this reach | N/A |
| | Basin_id | ID of the lake-network basin this reach belongs to | N/A |
| **Lake-network basins**: Entire drainage basins from the headwater to the ocean or inland sinks, defined by each of the inter-lake reach networks. Lakes in each lake basin are topologically related. File name: Basins_pfaf_xx | Basin_id | Lake-network basin ID | N/A |
| | Basin_type | Type of this lake-network basin: endorheic (draining to a terminal lake or an inland sink) or exorheic (draining to the ocean) | N/A |
| | Basin_area | Area of this lake-network basin | km² |

Earth System
Science
Data

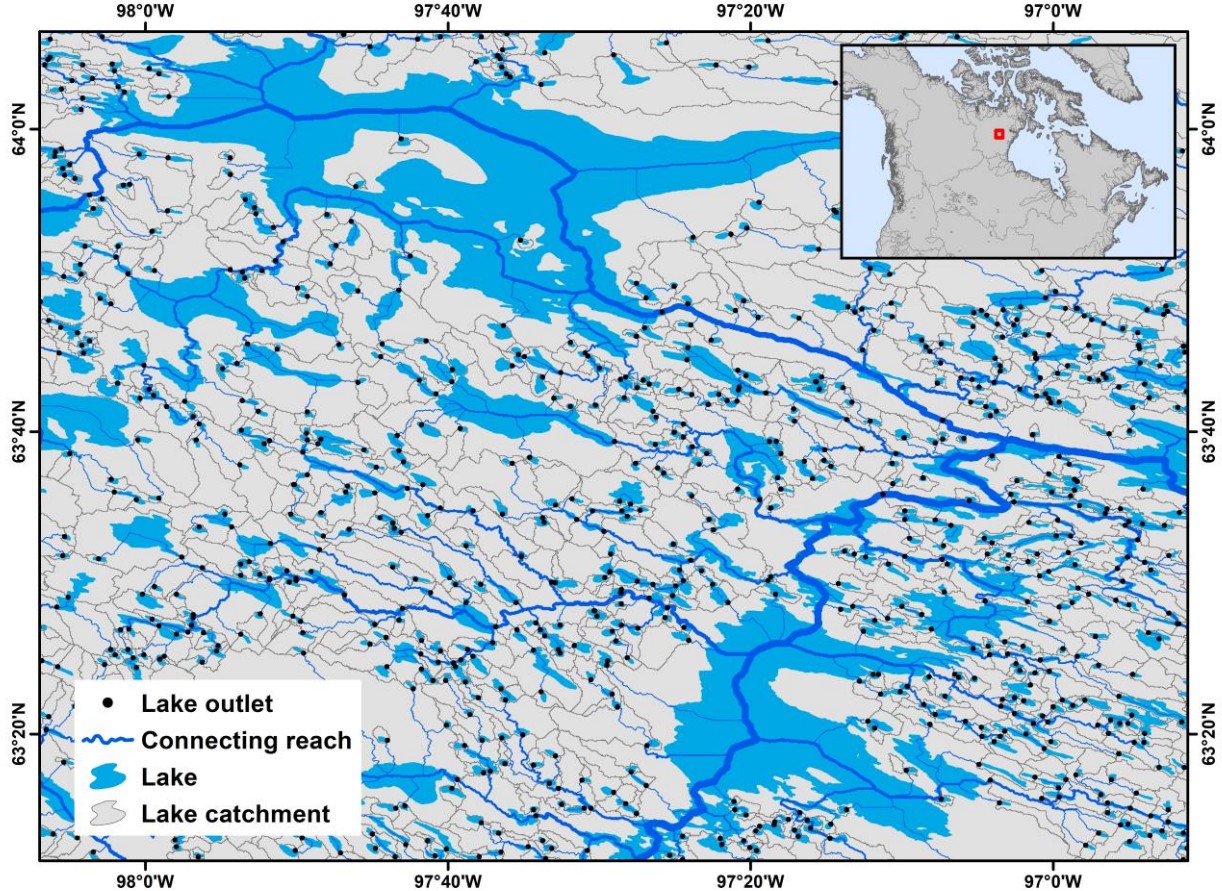

**Figure 1. An example of TopoCat in the Canadian Shield.** This example covers an area of ~6300 km$^2$ with ~850 lakes represented. Displayed features are lake outlets, inter-lake reaches (with width representing the abundance or count of the upstream lakes), lake unit catchments, and lake boundaries (lake boundaries are the same as HydroLAKES; Messager et al., 2016). All lakes in this region belong to the same lake-network basin shown in inset.

### 2.1 Lake boundaries

Lake boundaries in TopoCat are geometrically the same as the HydroLAKES lake polygons, except that the former includes new attributes informing lake drainage relations. About 99.95% of the ~1.4 million Hydro LAKES polygons were located within or on the MERIT Hydro boundary. These lakes were used to construct TopoCat (Fig. 2a-b). For consistency, we used the original HydroLAKES lake IDs ("Hylak_id") to index lakes and their associated topological attributes (Table 1). This allows users to link TopoCat features or attributes to HydroLAKES, as well as to other databases derived from HydroLAKES, such as the GLCP (Meyer et al., 2020), LakeATLAS (Lehner et al., 2022), the Global Lake Evaporation Volume (GLEV) dataset (Zhao et al., 2022), and the GLObal lakes Bathymetry dataset (GLOBathy) (Khazaei et al., 2022).

The attributes associated with each lake feature include the counts of the next (i.e., first or directly connected) upstream and downstream lakes ("U_lake_n" and "D_lake_n"), the ID(s) of the next downstream lake(s) ("D_hylak_id"), the count of all



upstream lakes which are connected in a cascade network from the headwater to this lake ("U_lak_ntot"), and the count of all
downstream lakes from this lake to the drainage sink or ocean ("D_lak_ntot"). To take into account lake bifurcation or
multifurcation, we also report the count of outlets for each lake ("Outlet_n"; see Section 2.2), with values larger than 1
indicating possible multifurcation (see the example in Fig. 3). The attributes also include the area of the entire lake catchment
("Cat_a_lake"), which is defined as the upstream drainage area from the headwater to the lake (note this is different from the
unit catchment areas; see Section 2.2). For a multifurcation lake, the lake catchment aggregates the upstream catchments
associated with all lake outlets (see Sections 2.2 and 2.3). More explanation for each of the attributes is given in Table 1.

TopoCat also provides the drainage type of each lake (hereafter "lake type") based on the position of the lake in relation to the
associated drainage network (Fig. 2a-b). Following the terminology in lake surface hydrologic position (Martin and Soranno,
2006; illustrated in Fig. 2a-b), our considered lake types include: (1) headwater, if the lake spatially concurs with the river
origin, (2) inflow-headwater, if the lake is fed by a river without any upstream lakes, (3) flow-through, if the lake is located on
the pathway of a river drained from an upstream lake, (4) terminal or endorheic, meaning that the lake is located at an inland
sink or terminal, (5) coastal, indicating the lake is immediately connected to the ocean (such as a lagoon) without any evident
outflow river , and (6) isolated (or seepage), if the lake is disconnected from the surficial river networks.

We acknowledge that the reliability of the configured lake type depends on the accuracy and completeness of the global river
networks, and some of the "isolated" lakes may actually be connected through non-channelized surface runoff and/or
subsurface flow (associated with surficial terrain). In addition, since lake-river connectivity can be intermittent, an "isolated"
lake may also be connected back to the network during high-flow conditions. For these reasons, we offer two a priori lake type
attributes to allow for more flexibility (Table 1). The first is "Lake_type," where lake drainage types were configured based
on the drainage paths connecting the entirety of global lakes (hereafter "inter-lake connecting reaches"), which we delineated
from the 3-arc-second MERIT Hydro hydrography dataset (Section 2.4). In other words, the inter-lake connecting reaches are
composed of a set of global drainage networks, where drainage density is determined by the observed spatial variability of
lake density. "Lake_type" ensures that all global lakes are topologically connected and eliminates any "isolated" and
"headwater inflow" lakes from lake drainage configuration. The number of identified headwater, flow-through, endorheic, and
coastal lakes as "Lake_type" are 57.3%, 42.5%, 0.1%, and 0.1% of the total lakes, respectively. The other attribute is
"Laktyp_mhv". Lake types in "Laktyp_mhv" attribute was configured based on MERIT Hydro–Vector (Lin et al., 2021), a
high-resolution global river network dataset which accounts for variable drainage density across the land surface (see Section
3.1 for more details). This attribute represents how global lakes are related to the perennial and intermittent rivers. About 4.5%
of the global lakes are categorized as "isolated", meaning that they are disconnected from any rivers in MERIT Hydro–Vector
(Fig. 2b). The drainage types for the other lakes are the same as those in attribute "Lake_type". The number of identified
isolated, headwater, inflow-headwater, flow-through, endorheic, and coastal lakes as "Laktyp_mhv" are 31.5%, 15.0%, 20.8%,
32.5%, 0.1%, and 0.1% of the total lakes, respectively.





**Figure 2. Global lakes and their drainage positions in TopoCat**. (a) Lake types ("Lake_type" attribute) configured based on the hydrologic positions in the inter-lake reaches (see section 2.4) of TopoCat. (b) Lake types ("Laktyp_mhv" attribute) configured based on the hydrologic position in global perennial and intermittent rivers provided in MERIT Hydro-Vector (Lin et al., 2021). Pie charts in both figures show the global lake area composition by drainage type. Here, lake boundaries are the same as HydroLAKES (Messager et al., 2016).



The pie chart in Fig. 2a shows the global lake area composition according to the lake drainage types based on the "Lake_type" attribute. Here, flow-through lakes cover the largest proportion of the global lake surface area (~67.1%), although only ~42.5% of the global lakes were identified as flow-through. Usually, flow-through lakes are larger in size than any of the upstream types (i.e., isolated, headwater, and inflow-headwater). Therefore, covering the largest lake surface area with relatively low

frequency was expected. Similarly, the pie chart in Fig. 2b shows the global lake area composition according to the lake drainage types based on "Laktyp_mhv". Considering that isolated lakes have the potential to be connected to the other lakes, 15.5% of the isolated, 16.3% of the headwater, and 12.7% of the inflow-headwater lakes from "Laktyp_mhv" become flow-through lakes in "Lake_type", leading to a 30.7% increase in the flow-through lake frequency. The dominant proportion for flow-through lakes further accentuates the intrinsic roles of lakes in the global drainage system and the necessity of building

this global-scale lake topology dataset. Endorheic or terminal lakes also cover a significant proportion of the global lake area (~21%), mainly due to large terminal saline lakes such as the Caspian Sea, the Great Salt Lake, and Lake Balkhash, as well as the cluster of terminal lakes across the endorheic Tibetan Plateau (Fig. 2a-b).

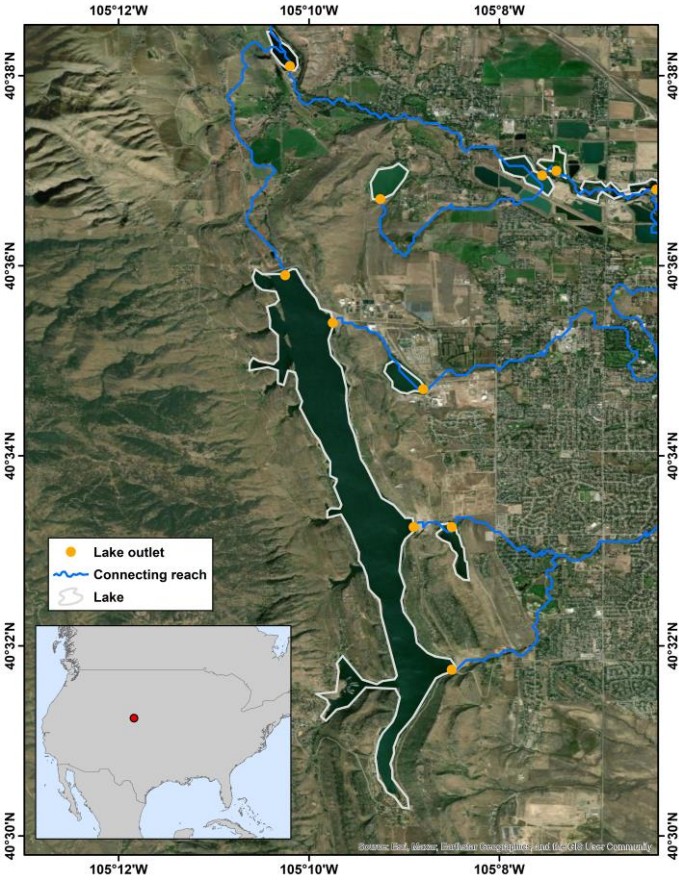

**Figure 3. An example of a multifurcation lake (Horsetooth Reservoir in Colorado, US) with four verified drainage outlets (lake**
**boundaries are the same as HydroLAKES; Messager et al., 2016, and map background from Esri, Maxar, Earthstar Geographics,**
**and the GIS User Community).**



## 2.2 Lake outlets

The outlet features, in principle, represent the locations of the maximum flow accumulation pouring from each lake to each of its next downstream lakes or drainage sink. If a lake is terminal, the outlet refers to the drainage sink. Although the majority
of lakes have one outlet only, lake bifurcation or even multifurcation does exist in both natural lakes and manmade reservoirs (see the example in Fig. 3). To provide a more realistic representation of global lake topology, we leveraged the detailed lake boundaries in HydroLAKES and the high-accuracy hydrography information in MERIT Hydro (see Section 3.1) to allow for possible multifurcation. The method is elaborated on in Section 4. However, it is important to note that only water body maps were used to develop the MERIT Hydro, without considering lake masks as an input. Therefore, in many lakes, flow directions
within the lake might be influenced by adjacent topography and causing multiple outlets. Since multifurcation was considered, the count of outlets in TopoCat is larger than the count of lakes. We identified 1,459,201 outlets for 1,426,967 lakes, where 29,190 lakes (~2% of the global lakes) drain to multiple destinations (Fig. 4).

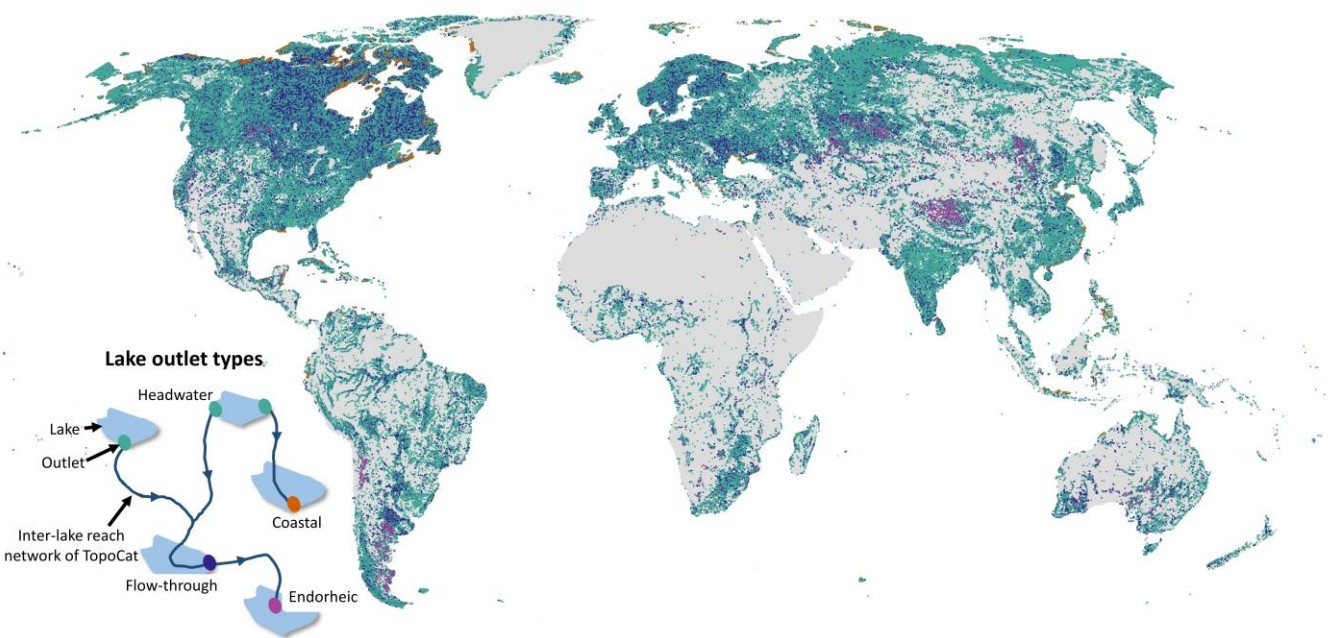

**Figure 4. Global map of lake outlets and drainage types in TopoCat.** Outlet types are based on drainage positions in the inter-lake reach network. The schematic diagram shows the outlet type definitions.

The outlet attributes contain the topology information detailing how the outlets, lakes, and catchments are hydrologically connected to each other. They include the unique ID of the outlet ("Outlet_id"), the associated HydroLAKES lake ID ("Hylak_id"), and the drainage relationship with other outlets and lakes, such as the IDs of the next downstream outlets ("D_out_id"), the IDs of the next downstream lakes ("D_hylak_id"), and the count of the next upstream lakes ("U_lake_n").





The attributes also contain the area of the unit catchment associated with each outlet ("Cat_area"). Here a unit catchment (also see Section 2.3) is defined as the drainage area between any two cascading outlets (i.e., from an outlet to its next upstream outlet), or the drainage area from the headwater to this outlet if there is no lake upstream. In addition, we also report the area of the entire upstream catchment ("Cat_a_tot"), defined as the drainage area from the headwater to this outlet, the count of lakes within the entire upstream catchment ("U_lak_ntot"), and the count of lakes within the entire downstream catchment to

the inland sink or ocean ("D_lake_ntot").

Similar to lake types, the type of each lake outlet ("Outlet_type") was assigned based on the drainage position in our delineated inter-lake reach network (Fig. 4). For lakes with only one outlet, the outlet type equals the corresponding lake type ("Lake_type"). For multifurcation lakes with disagreeing outlet types, the outlet type associated with the most downstream position took precedence in representing the lake type. For instance, a multi-outlet lake might have a headwater outlet and a

flow-through outlet at the same time, which are flowing in two different directions. Here, the flow-through outlet has at least one upstream lake, and the headwater outlet has no upstream lake. This mainly occurs due to the limitation of the D8 flow-direction, which cannot depict flow-bifurcation. In this case, the lake was treated to be on the drainage pathway and thus assigned to be a "flow-through".

We also provided useful proximity information that measures how global lakes are close to one another in terms of drainage

distance, i.e., the geodesic distance along our delineated inter-lake reaches (Section 2.4). These attributes include "D_distance," defining the drainage distance from each lake outlet to the next downstream outlet; "D_dist_sink," measuring the cumulative drainage distance from the lake outlet to the inland sink or the ocean; and the maximum, minimum, and mean drainage distances to the next upstream outlets ("U_dist_max", "U_dist_min", "U_dist_avg").

## 2.3 Unit catchments

The unit catchment features depict the spatial boundaries of unit catchments (see definition in Section 2.2) associated with each of the lake outlets. Note that unit catchments are not based on one-lake to one-catchment relationship but based on one-outlet to one-catchment relationship. This was to ensure that the catchment boundaries were given at a spatial detail consistent with the lake outlets or drainage destinations. This way, the count of unit catchments is equal to the count of lake outlets, and bifurcation or multifurcation lakes have multiple unit catchments.

Attributes of each unit catchment include the ID of the associated outlet ("Outlet_id"), the ID of the associated lake ("Hylak_id"), the IDs of the next downstream outlet and lake, respectively ("D_out_id" and "D_hylak_id"), and the area of this unit catchment ("Cat_area"). We also reported the drainage type of each unit catchment in relation to the inter-lake reach network ("Cat_type"), which is identical to the drainage type of the associated outlet ("Outlet_type"). The pie chart in Fig. 5 summarizes the composition of global lake unit catchment areas in terms of their drainage types, where flow-through

catchments cover the largest proportion (~79%) in area, followed by headwater catchments and terminal catchments. It is important to note that a terminal unit catchment is not always equivalent to an endorheic basin. The former refers to the unit catchment of a terminal lake, which is the drainage area between the sink and the next upstream lake(s), whereas the latter



refers to the entire landlocked drainage basin from the headwater to the sink (i.e., the cases where "Basin_type" = "endorheic" in "Lake-network basins" features).

It is worth noting that although the unit catchments were offered at the outlet level, when needed, users can easily dissolve them based on their lake IDs ("Hylak_ID") to form the local catchment boundaries for each lake (i.e., the drainage area from the upstream lake(s), if any, to this lake). In total, the delineated unit catchments in TopoCat cover about 77.5 million $km^2$, which is about 26.4 times of the total lake area and about 57% of the Earth's land mass excluding the Antarctic (Fig. 5).

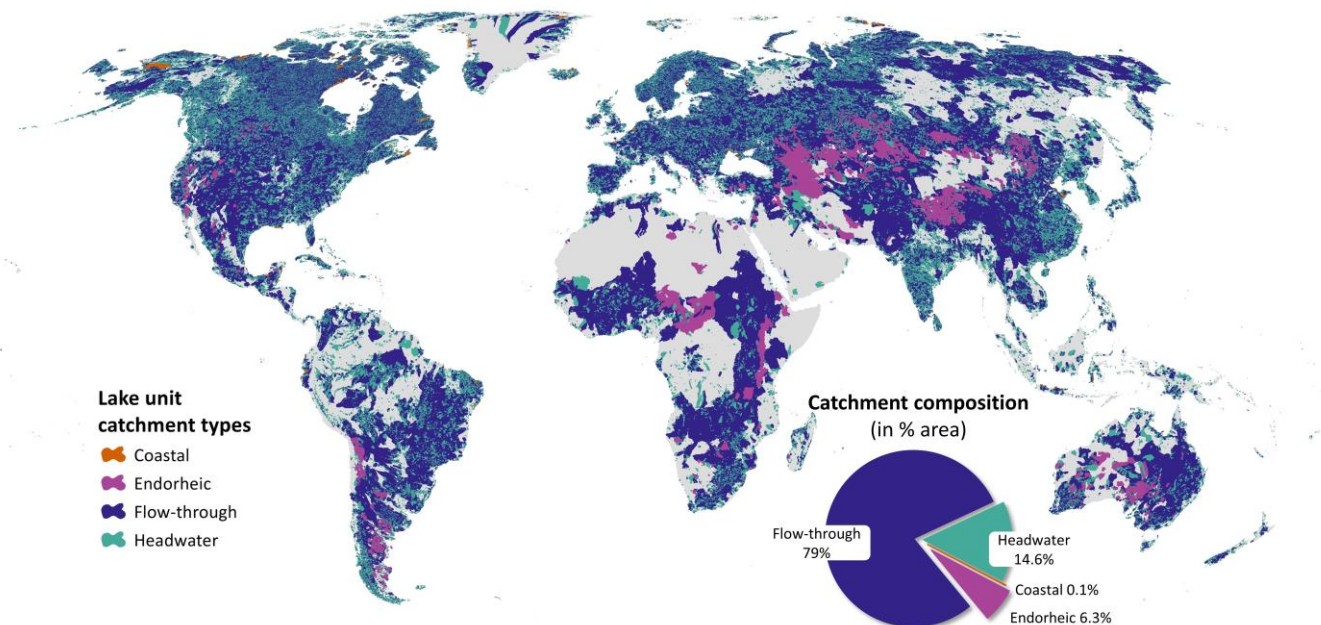

**Figure 5. Global map of lake unit catchments in TopoCat.** The catchment boundaries define the drainage areas between cascading lake outlets (i.e., one catchment for each lake outlet), and when there is no lake further upstream, the catchment defines the drainage area from the headwater to the lake outlet. The pie chart shows the composition of global lake catchment area by outlet drainage type.

### 2.4 Inter-lake reaches

The inter-lake reach features contain a set of detailed reach networks that tie any lake in the world to its upstream and
downstream lakes (if any) through the outlets. Together with the topology attributes (Table 1), each inter-lake reach network depicts how the lakes in the headwater drain to other lakes downstream in a cascade pattern, which eventually reaches a drainage terminus, i.e., an inland sink or the ocean. Because these networks are determined by lake distribution, their drainage density varies with lake density across the continental surface. Such inter-lake networks can be different from other river datasets such as MERIT Hydro–Vector, which were not tailored for lake-determined drainage density. More specifically, each
reach in an inter-lake network starts from the outlet of one lake and ends with the outlet of the next downstream lake, but along the way, the reach is often segmented further by the confluences with tributary reaches from other lakes. If a lake is not coastal, endorheic, and does not have a downstream lake, we kept the outflowing reach from this lake all the way to the ocean or inland

sink. In other words, a reach network always stretches from the most upstream lakes to the drainage terminus even when there
is no terminal lake. About 3 million inter-lake reaches were generated among the 1.46 million outlets (Fig. 6). The total length
of these reaches is about 10 million km.

Main attributes of each reach include the unique reach ID ("Reach_id"), the IDs of the connected upstream outlet and lake,
respectively ("U_out_id" and "U_hylak_id"), the count of all upstream lakes from the headwater ("U_lak_ntot"), the ID of the
next downstream reach ("D_reach_id"), the IDs of the connected downstream outlet and lake, respectively ("D_out_id" and
"D_hylak_id"), the geodesic distance and slope of the reach ("Rch_length" and "Rch_slope"), and the Strahler stream order
(Strahler, 1957) of the reach based on the inter-lake network ("Rch_order"). Figure 6 shows the global distribution of our
delineated inter-lake reach networks, where different colors illustrate the total abundance (count) of lakes upstream of each
reach (as in the attribute "U_lak_ntot"). For example, the abundance of upstream lakes per reach in some of the most lake-rich
regions, such as the Canadian Shield and Scandinavia, can be several orders of magnitude higher than the rest of the world.

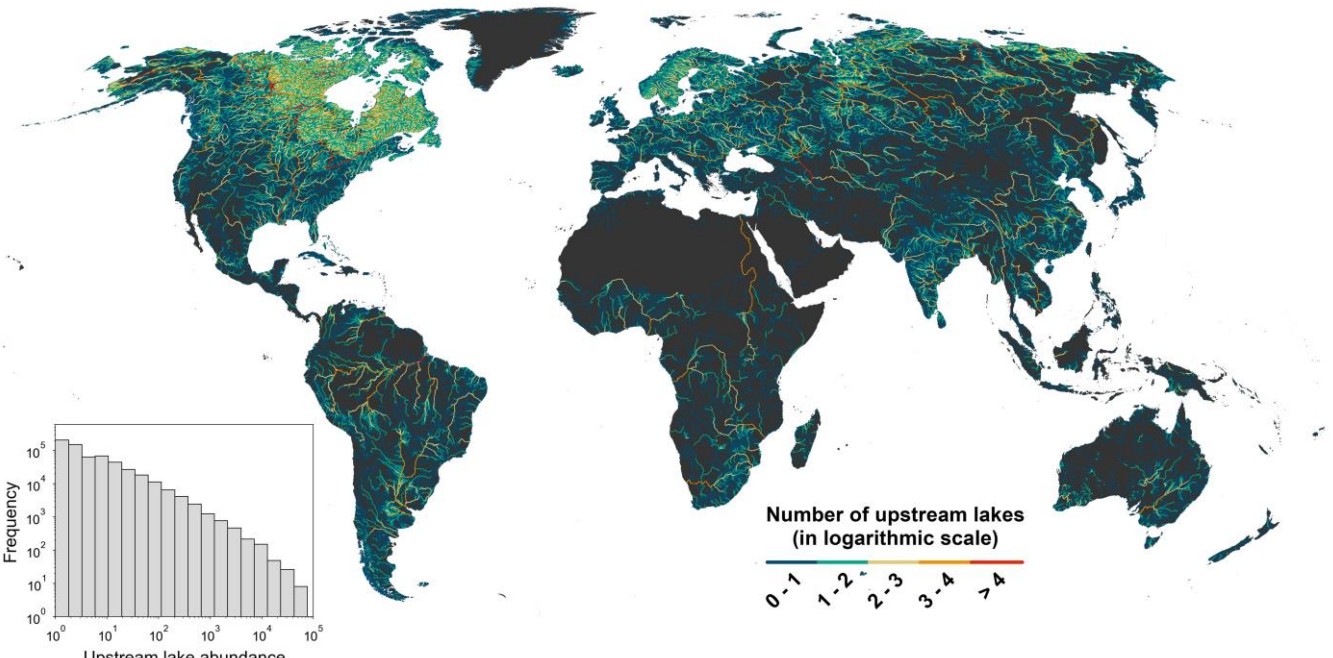

**Figure 6. Global map of inter-lake reaches in TopoCat.** Reach colors illustrate the accumulative lake abundance or count upstream to
each inter-lake reach. The histogram on the lower-left corner shows the frequency of upstream lake abundance.

As previously noted, the inter-lake reach networks aim to define the potential drainage paths connecting global lakes and
reservoirs. These reaches are not always perennial or intermittent rivers, but can be highly temporally dynamic, non-perennial,
and non-channelized. Similar to the lake type assignment, we here give a rough inference of the type of each reach based on
its relationship with the river channels in MERIT Hydro-Vector. We reported the proportion of each inter-lake reach
("Rchint_mhv") overlapped by river channels in MERIT Hydro-Vector. The overlapping proportion indicates: "probably
river" if an inter-lake reach is completely contained by the river network in MERIT Hydro-Vector (i.e., "Rchint_mhv" =

100%), "partially river" if the inter-lake reach partially overlaps MERIT Hydro-Vector (i.e., "Rchint_mhv" < 100% and > 0%), and "other drainage path" if the inter-lake reach is beyond the spatial extent of the river network in MERIT Hydro-Vector

(i.e., "Rchint_mhv" = 0%).

**2.5 Lake-network basins**

For user convenience, we provided another feature file "Lake-network basins". Each lake-network basin, in principle, defines the entire drainage area of an inter-lake reach network, which stretches from the headwater to an inland sink or the ocean. Nevertheless, our inter-lake reaches were constructed based on lake outlets (refer back to Section 2.4). If a multifurcation lake

infringes more than one reach networks that drain to different termini, these networks were considered related to each other through this shared lake, and their drainage areas were merged into one lake-network basin.

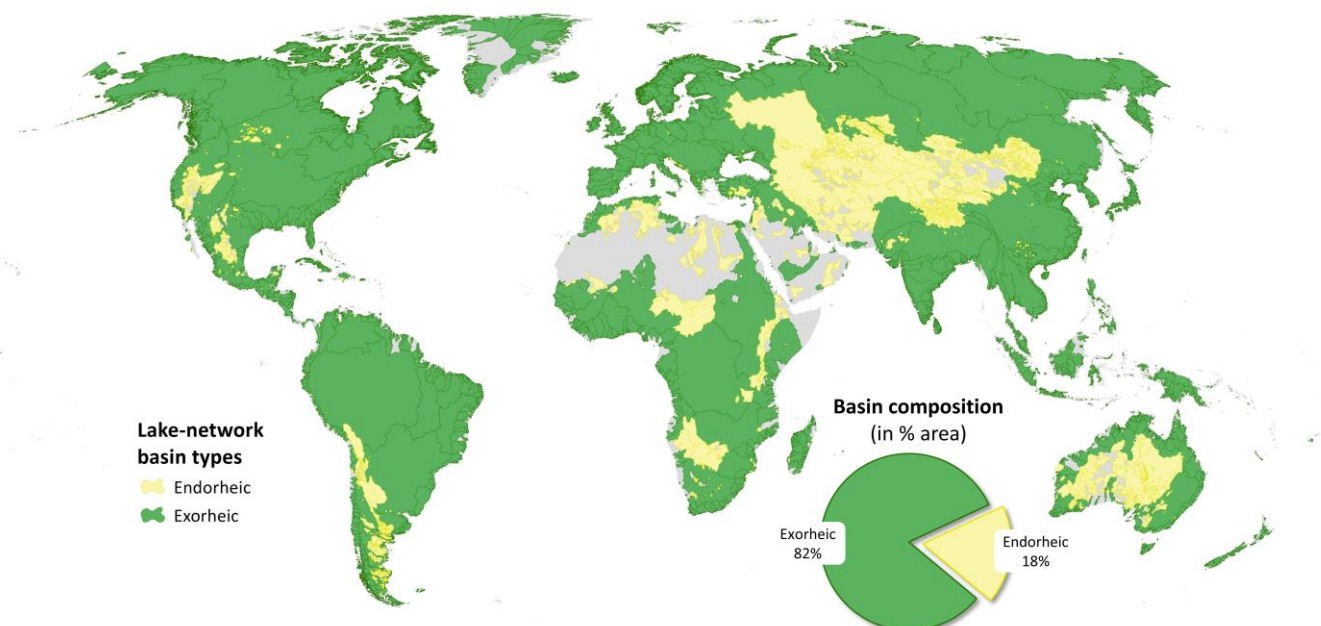

**Figure 7. Global map of lake-network basins in TopoCat.** A lake-network basin is defined as the entire drainage area from the headwater to the ocean or the inland sink, which contains a hydrologically independent inter-lake network. Gray indicates basins having no lake

presence. The pie chart illustrates the composition of global lake basin areas by their drainage type (endorheic and exorheic).

This logic resulted in 47,340 lake-network basins in the world as shown in Fig. 7. Among them, endorheic basins (as indicated by the "Basin_type" attribute) account for 5.1% by count and 18% by area of all lake-network basins. These endorheic basins cover ~15.4% of global surface excluding Antarctica. This proportion is smaller than the conventional size of endorheic basins, which is about a fifth of the global landmass excluding Antarctica (Wang et al., 2018; Wang, 2020). This is because a lake-

network basin delineated here must contain at least one lake, and some of the conventional endorheic basins, such as with extreme aridity in the Sahara Desert, the Arabian Desert, and the Gobi Desert, are arheic and have no lake presence (according to HydroLAKES), which led to their exclusion from our delineated lake-network basins. But overall, the spatial pattern of the

lake-network basins is consistent with the river basins, signifying again that lakes are an integrated component of the global river system. Importantly, these lake-network basins define how the millions of global lakes are partitioned by their drainage

dependence, which can potentially help users gauge or optimize parallel processing (e.g., for global lake modeling and coupled lake-river modeling). For this reason, we also assigned the ID of the affiliated lake-network basin ("Basin_id") to each of the other TopoCat features.

## 3 Input and validation data

Lake-TopoCat v1.0 was developed based on HydroLAKES v1.0 (Messager et al., 2016) primarily using the 3-arc-second

MERIT Hydro (v1.0.1) hydrography dataset (Yamazaki et al., 2019), together with the river flowlines in MERIT Hydro-Vector (Lin et al., 2021) to assist in lake and reach type configurations. The developed lake topology and catchments were validated against: (1) the high-quality National Hydrography Dataset (NHD) (McKay et al., 2012) in part of the US; and (2) the quality-controlled lake basin boundaries on the endorheic Tibetan Plateau (Liu et al., 2020; Liu et al., 2021). More details of each of the input and validation datasets are given below.

### 3.1 Lake mask

Constructing global lake topology requires two essential types of data: a global lake inventory with detailed lake boundaries; and a DEM or hydrography dataset depicting accurate drainage directions. For the former, we selected the widely used HydroLAKES, which contains about 1.4 million lake polygons larger than 10 ha (0.1 km$^2$) compiled from various digital and remote sensing products (Messager et al., 2016). HydroLAKES is, to our knowledge, one of the most comprehensive global-

scale lake and reservoir inventories publicly available. Associated with each lake boundary, HydroLAKES also provides the outlet point and a number of attributes such as lake mean depth, lake volume, and catchment area. The lake outlet points and catchment areas in HydroLAKES were derived from the 15 arc-second (~500 m at the equator) HydroSHEDS database (Lehner et al., 2008) without consideration of bifurcation, and were later used for comparison with the outlets and lake catchment areas in TopoCat (see Section 5.1 for details).

### 3.2 Hydrography

We considered two global hydrography datasets for building lake drainage topology: HydroSHEDS (Lehner et al., 2008) and MERIT Hydro (Yamazaki et al., 2019). The HydroSHEDS hydrography database is available at three different resolutions. Limited by the spatial coverage of SRTM DEM (Farr et al., 2007), the highest-resolution HydroSHEDS v1.1 at 3 arc-second (~90 m at the equator) is missing the landmass north to 60˚ N, where 50.5% of the global lakes (by count) are distributed.

Although the 15 arc-second version covers the pan-Arctic region above 60˚ N, the accuracy of the hydrography is significantly inferior due to a poorer quality of the underlying HYDRO1k DEM (EROS Center, 2018). On the other hand, MERIT Hydro v1.0.1 was developed consistently based on the 3 arc-second, high-accuracy MERIT DEM (Yamazaki et al., 2017) with a

complete coverage of the global landmass (excluding Antarctica). Compared with SRTM DEM, MERIT DEM eliminated major error components including absolute bias, stripe noise, speckle noise, and tree canopy height, which often distort surface

drainage directions and fragment drainage paths. The production of MERIT Hydro also employed a "stream-burning" technique on MERIT DEM. This technique applied multi-source waterbody datasets to derive flow directions that were more realistically aligned with the observed drainage networks. Small dummy depressions were then removed by an inland basin detection algorithm to further smooth the drainage continuity. For these reasons, we selected MERIT Hydro to configure lake outlets, catchments, and interconnecting drainage paths. As one of the supplementary layers, MERIT Hydro also produced a

hydrologically adjusted DEM, where elevations were ensured to comply with the condition that downstream should not be higher than upstream. This hydrologically adjusted DEM was used to calculate the drainage gradient (slope) of each inter-lake reach.

Furthermore, we used the vector-based global river network dataset, MERIT Hydro-Vector (Lin et al., 2021), to assign the type of each lake based on its drainage position in the network (attribute "Laktyp_mhv"). In this dataset, the spatial variability

of drainage density was considered when the networks were delineated from the original MERIT Hydro. A machine learning technique was employed to optimize the drainage density per watershed (with a flexible channelization threshold down to 1 $km^2$) in relationship with various hydroclimate factors. About 58 million river reaches were delineated, with a total length of ~75 million km. It is important to note that the point of MERIT Hydro-Vector was not to simply vectorize MERIT Hydro to the finest spatial detail (as something done by Hydrography90m (Amatulli et al., 2022)), but to simulate spatially variable river

networks so that the vectorized channels resemble the actual densities of perennial and/or intermittent rivers in the world. Although the temporal dynamics of channel heads due to climate variability were not accounted for and the resolvable headwater sizes are constrained by the spatial resolution of MERIT DEM, we considered the flowlines in MERIT Hydro-Vector as one of the best available global representations of perennial and intermittent river networks.

**3.3 Validation**

Detailed and accurate information of lake topology and basins is lacking on a global scale, so we benchmarked TopoCat against two high-quality regional hydrographical geofabrics. NHDPlusV2.1 (McKay et al., 2012), a value-added version of the 1:100,000 NHD dataset (USGS 2001), was used to validate our derived lake catchment areas in the continental US, and lake topology in part of the Prairie Pothole Region in the US, one of the world's most lake-dense areas. NHDPlusV2.1 contains catchment boundaries for NHD on-network waterbodies and their drainage topology based on flow directions. Here, on-

network waterbodies are those which intersect the NHD flowline network. For the waterbodies off the NHD network, we obtained catchments and drainage topology from the LakeCat dataset (Hill et al., 2018). Together, these datasets represent the most up-to-date and comprehensive lake catchment and drainage topology information for the US.

In addition, we used the watershed boundaries on the Inner Tibetan Plateau produced by Liu et al. (2021) to validate the endorheic lake catchments for the same region in TopoCat. The Inner Tibetan Plateau hosts the world's largest cluster of

terminal lakes. Delineating lake catchments in arid and endorheic basins can be challenging for at least two reasons. First,

small pseudo depressions due to errors or noise in the elevation data often coexist with actual inland basins, and eliminating them from the hydrography data is difficult (Yamazaki et al., 2019). Second, many terminal lakes on the Tibetan Plateau are rapidly expanding due to a wetting and warming climate (Chen et al., 2022b; Wang et al., 2018; Yao et al., 2018). Such lake dynamics have led to ongoing drainage re-organizations such as basin merging and annexation, meaning that the catchments

derived from a static lake mask, (such as HydroLAKES) are prone to boundary migration through time. Despite the availability of other inland lake catchment data for this region (e.g., Yan et al., 2019), the dataset from Liu et al. (2021) is, to our knowledge, the only one that considered the ongoing drainage changes on the Tibetan Plateau due to lake inundation dynamics. They used a lake-oriented approach (Liu et al., 2020), together with the calibration of multi-temporal lake mappings, to refine the basin boundaries initially derived from MERIT DEM. The dataset offers the endorheic lake catchments for the years 2000 and 2018

(hereafter "Liu21 endorheic basins 2000 and 2018"), with a total catchment number of 434 and 421, respectively.

## 4 Methodology and algorithm

The TopoCat constructing algorithm is illustrated by the schematic diagram in Fig. 8. In brief, the algorithm started with an iterative process that used MERIT Hydro's flow accumulation layer to identify all possible outlets per lake (i.e., pour points to all possible outbound directions from this lake) and each of their associated unit catchments. The connecting reaches among

the lake outlets were then delineated using the flow direction layer. The majority of attributes for lake outlets, unit catchments, and inter-lake reaches (Table 1) were then generated using the hydrography, hydrologically adjusted DEM, and the spatial relationships among the delineated features. This resulted in the "preliminary TopoCat" database, which consists of the feature geometries and topological attributes in the finest spatial detail possible, based on all identified lake outlets. For user convenience, we also release the preliminary TopoCat (with the affix "_prelim" in the file names) together with the final

cleaned-up version (as explained in Section 2). The definitions of attributes are consistent in both versions (Table 1).

Next, the preliminary TopoCat was simplified based on the principle of "one outlet per drainage destination (i.e., downstream lake) per lake", which formed the final cleaned-up version of TopoCat. Conceptually, if a lake has multiple preliminary outlets draining to the same lake, a representative outlet was selected to be the one with the maximum flow accumulation. Accordingly, all preliminary catchments of a lake that drain to the same destination were dissolved to one final unit catchment, and the

subset of the preliminary reaches connecting the representative lake outlets were extracted as the final reaches. The existing attributes were then re-calculated according to the updated network. The lake-network basins were delineated using the terminal points of all the available inter-lake reach networks. In the last step, a few additional lake outlet attributes were populated and assigned to the associated lake polygons, and MERIT Hydro-Vector was used to determine lake types ("Laktyp_mhv") according to their hydrological position in the river networks.





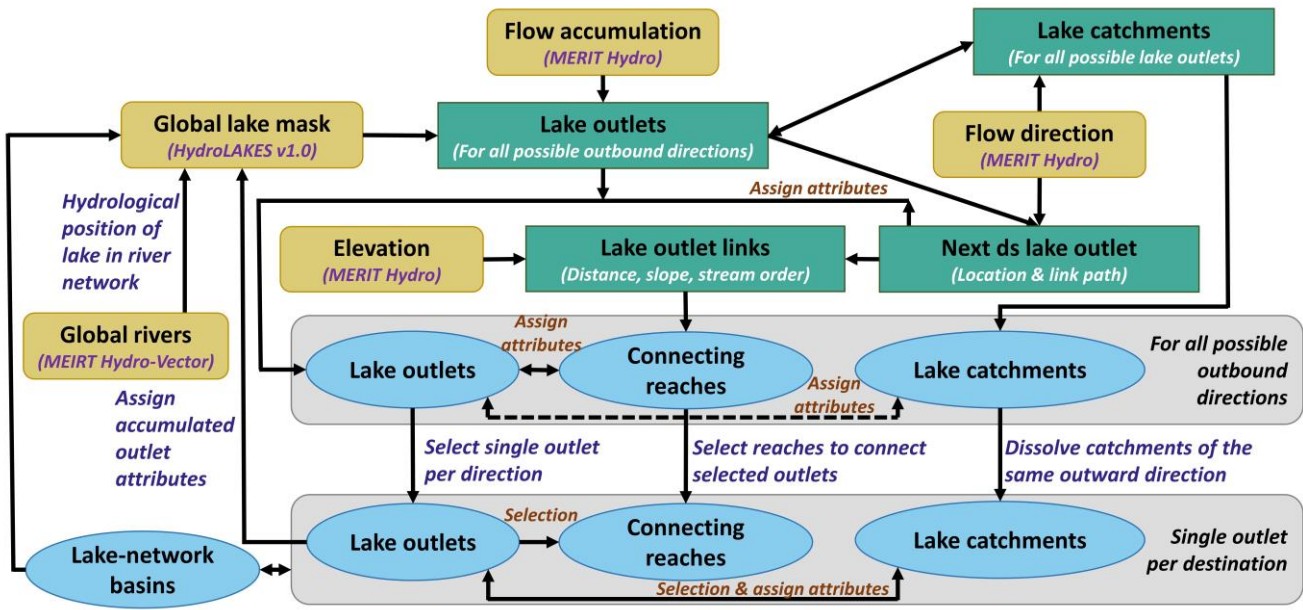


**Figure 8. Schematic diagram of the algorithm used to construct TopoCat.** Rounded corner rectangles, rectangles, and oval shapes represent input data, intermediate products, and final TopoCat products, respectively.

To enable parallel processing, we partitioned the MERIT Hydro dataset into multiple Pfafstetter level-2 basins. A total number

of 61 Pfafstetter level-2 basins were obtained from HydroBASINS (Lehner and Grill, 2013), which were originally generated

from the HydroSHEDS hydrography data (Lehner et al., 2008). Lin et al. (2021) redefined these level-2 basin boundaries to

make them compatible with MERIT Hydro. This pre-processing, however, only considered watersheds larger than 25 km$^2$, so

we further updated the 61 level-2 basins by annexing any smaller watersheds to the nearest major watershed. In addition, a

number of islands and archipelagos scattered across the Pacific, Atlantic, and Indian Oceans were covered by MERIT Hydro

but excluded from the level-2 basins. We grouped these islands into another seven clusters to form a total of 68 Pfafstetter

level-2 basins or regions (Fig. 9). These 68 regions cover all pixels in MERIT Hydro and were used to define the subset

domains of our TopoCat product. For user convenience, we provided the polygon boundaries of these 68 Pfafstetter regions in

the file named "Pfaf2_regions".





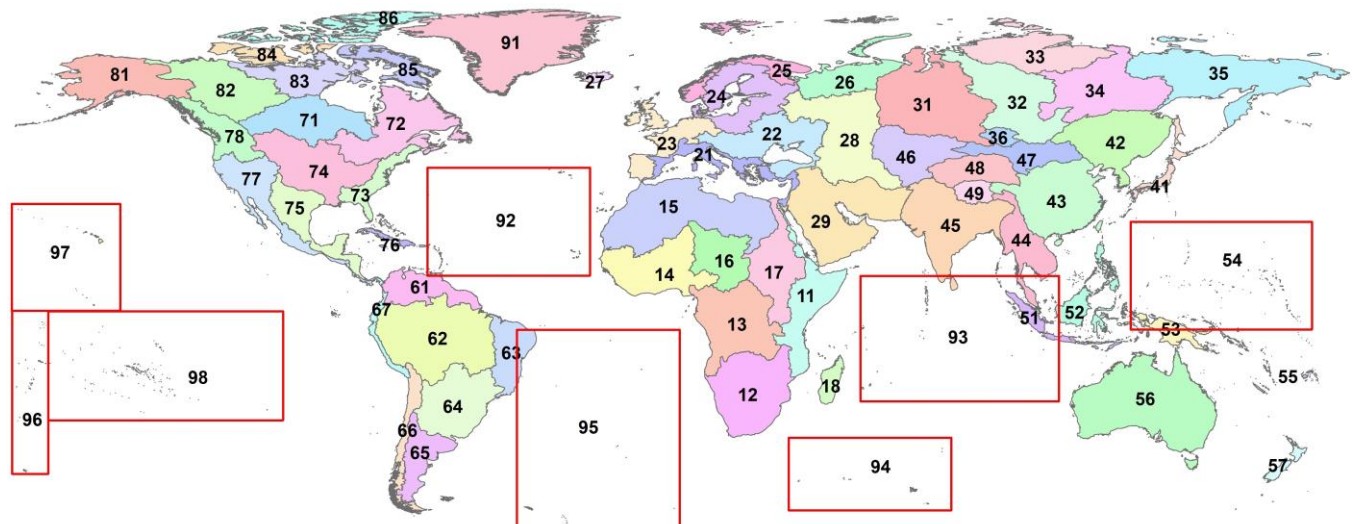

**Figure 9. A total of 68 partitioning regions (Pfafstetter level-2 basins) in the world for TopoCat production and subset organization.**

The above-mentioned algorithm was implemented in six specific steps. All steps were automated by python scripts with the ArcPy geoprocessing package, except step 4 which required manual quality control for the identified terminal outlets (see Section 4.4). This manual quality control took approximately 25–30 expert-hours, and much of it was a one-time process that will require no replication even if the input lake mask is updated. We describe each of the steps in the sections below.

### 4.1 Delineating preliminary lake outlets and unit catchments

Although HydroLAKES provides lake outlet points derived from the 15 arc-second HydroSHEDS data, we regenerated the lake outlets using the higher-resolution (5 arc-second) MERIT Hydro dataset. Different from the outlets available in HydroLAKES, we were not constrained by the "one-outlet for one-lake" assumption in order to capture possible lake multifurcation. However, we had to assume that multifurcation can only occur at the lake level, not the outlet level, because pixel-based bifurcation was not assumed in MERIT Hydro. Typically, the outlet location of a lake can be identified by

intersecting the lake shoreline with the flow direction layer and then locating the intersecting cell(s) flowing outward from the lake (Tokuda et al., 2021). However, there is no guarantee that the lake outlets always fall exactly at the shoreline cells. An example is an endorheic lake, where the drainage outlet (the sink in this case) is located inside the lake boundary. Other examples are some of the lagoon polygons in HydroLAKES, which can overshoot the terrestrial boundary of MERIT Hydro due to resolution or precision inconsistency between the lake and hydrography datasets. To address these issues, we adopted

a flow-accumulation-based approach, which identified the lake outlet as the location of maximal flow accumulation within the lake boundary.

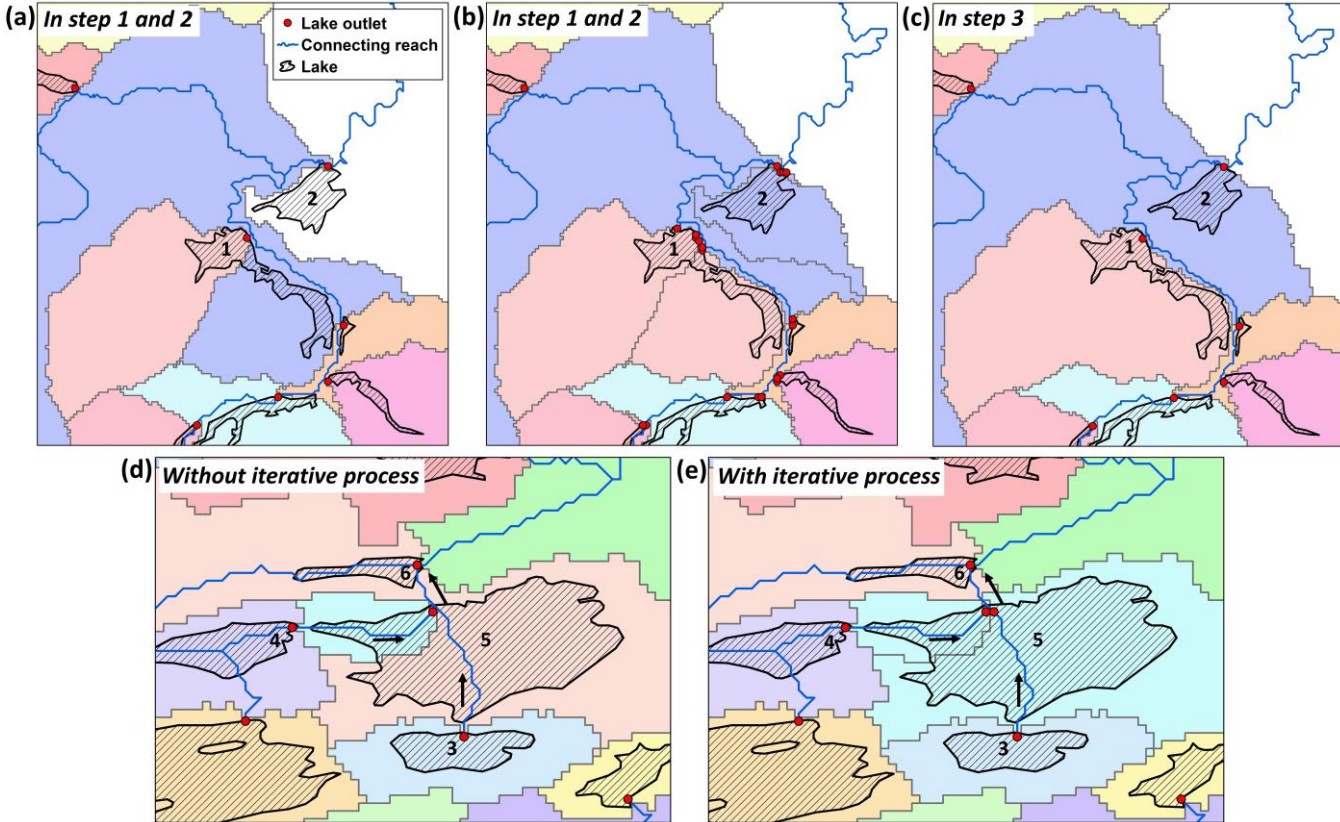

**Figure 10. Schematics of the iterative process for outlet and catchment delineations (upper panel) and the improvement in lake connectivity and catchment completeness due to this iterative process (lower panel).** Lakes are shown as striped polygons, lake unit catchments as color-filled polygons, lake outlets as red dots, and inter-lake reaches as blue lines. (a) Delineated primary lake outlets (i.e., no iteration) with their associated unit catchments and connecting reaches. Note that lake 1 and lake 2 are not contained by their unit catchments. (b) Delineated multiple preliminary outlets and their associated unit catchments through the iterative process. (c) The final selected outlets with their dissolved unit catchments and connecting reaches (described in Section 4.3). Lake 1 and lake 2 are now contained by their own unit catchments. (d) Lake topology and unit catchments generated by the non-iterative process. Lake 3 drains to lake 6 without intersecting the outlet of lake 5, and the unit catchment of lake 5 appears incomplete. (e) Lake topology and unit catchments delineated by the iterative process, with improved inter-lake drainage connectivity and unit catchment completeness. Here, both outlets of lake 5 have been maintained in the final TopoCat to conserve the topological connectivity between lake 3, 5 and 6. Here, lake boundaries are the same as HydroLAKES (Messager et al., 2016).

The flow-accumulation-based approach was employed iteratively until all possible outlets for each lake were identified. As illustrated in Fig. 10a, the cell where flow accumulation is the maximum within the lake boundary was first identified as the primary outlet, and the unit catchment associated with the primary outlet (hereafter "primary catchment") was delineated using flow direction. Logically speaking, the drainage basin of a lake should completely contain the lake boundary. If the primary catchment covers only part of the lake polygon, we assumed that possible multifurcation may occur to this lake. In this case, the second iteration was triggered to identify the next maximum-flow outlet (i.e., the secondary outlet) from the lake cells outside of the primary catchment, and the unit catchment associated with the secondary outlet was delineated (Fig. 10b). The iteration was repeated until the entire lake area was covered by its unit catchment(s). The iteration sequence, which corresponds



to a descending order of the outlet flow accumulation, was assigned to each outlet to keep track of the multiple outlets of the same lake. Locations of all possible lake outlets and their associated unit catchments were delineated through this iterative approach. Each outlet and the associated unit catchment were labeled with the same unique outlet ID, and associated lake ID.

We termed them as "preliminary outlets and unit catchments" to make a distinction from the representative outlets and their associated unit catchments in our final lake topology (see Section 4.4).

Despite topological redundancy, there are multiple benefits to retrieving all preliminary outlets using the above-described iterative process. First, this process was a necessary preparation for capturing lake multifurcation (see the example in Fig. 3). Second, identifying all lake outlets ensured a proper portrayal of the complete lake catchment (Fig. 10b-c). Third, it also

ensured that the lake drainage topology was consistently configured. In the example of Fig. 10e, both lake 3 and lake 4 first drain to lake 5 which then drains to lake 6. Without the iteration, the secondary outlet of lake 5 connecting lake 3 and lake 6 would have been missing, meaning that in the configured topology, the directly connected downstream lake from lake 3 would have been misidentified as lake 6.

**4.2 Building preliminary lake topology and inter-lake reaches**

Lake topology defines how different lakes are hydrologically connected to each other. In this step, such topological information, together with the inter-lake reach networks, was established using the preliminary lake outlets and MERIT Hydro. Given each preliminary outlet, the next downstream outlet was traced by routing from this outlet pixel based on the MERIT Hydro flow directions. The identified downstream outlet ID and its corresponding lake ID ("D_out_id" and "D_hylak_id") were saved in the attributes of both preliminary outlet and catchment features.

During this routing process, the drainage path connecting the preliminary outlets was delineated pixel by pixel until the path reached an inland sink or the ocean. The traced paths were converted into vector format to create an inter-lake reach network, where individual reaches were segmented by lake outlets, drainage termini (sinks or the ocean), and the confluences of the reaches (Fig. 10b). For each reach, the next downstream reach ID ("D_reach_id") was also configured to depict the drainage topology of this inter-lake reach network. Several other attributes of the reaches, such as the next upstream outlet ID

("U_out_id"), the next downstream outlet ID ("D_out_id"), and their associated lake IDs ("U_hylak_id" and "D_hylak_id"), were obtained through spatial relationships with the outlet features. The elevations of the start and end nodes ("Start_elv" and "End_elv"), as well as the geodesic length and slope of each reach ("Rch_length" and "Rch_slope"), were derived from the hydrologically adjusted MERIT DEM.

A few more attributes of the preliminary lake outlets and unit catchments were populated from their feature geometries and

the attributes of the inter-lake reaches. The area of each unit catchment ("Cat_area") was calculated and then linked to the associated lake outlet through the common outlet ID ("Outlet_id"). The accumulative total catchment area from the headwater to each outlet ("Cat_a_tot") was aggregated from the areas of individual unit catchments upstream to this outlet based on the configured outlet topology. The drainage distance and the slope to the next downstream outlet or sink ("D_distance" and "D_slope") were calculated using the topology and attributes of the inter-lake reaches.



### 4.3 Simplifying drainage topology and assigning outlet drainage types

In this step, redundancy in the preliminary topology was reduced to only retain the unique drainage destinations from each lake. Specifically, all preliminary outlets of a lake were first grouped based on their next downstream lake IDs. A lake possessing multiple outlet groups, i.e., different downstream lakes, indicate possible multifurcation. Within each group, the outlet having the highest flow accumulation value was selected as the final representative outlet for this drainage destination (Fig. 10c). If multiple outlets within a group have the same highest flow accumulation value, the outlet with the shortest drainage distance to the next downstream lake was selected. If this still did not lead to a single solution, an outlet in the above-mentioned scenario was randomly selected. Limited by the input data quality, there are occasional cases where part of a lake drains back to itself through another lake (or several other lakes), forming a pseudo drainage loop. Such drainage loops and the associated outlets were considered as artifacts and thus automatically removed, if removal of such loops did not hamper the connectivity with other lakes. A few of the attributes of this final outlet, such as the downstream outlet ID ("Outlet_id"), drainage distance and slope to the downstream outlet ("D_distance" and "D_slope"), were reconfigured to reflect this simplification process.

Similar to the preliminary outlets, the preliminary unit catchments of a lake were also grouped by their unique drainage destinations. Each group of preliminary catchments was dissolved to a single unit catchment (Fig. 10c), and the attributes of this dissolved unit catchment, such as the outlet ID ("Outlet_id") and the next downstream outlet ID ("D_out_id"), were updated according to the attribute information of the associated final outlet (which shares the same drainage destination). The area of the dissolved unit catchment ("Cat_area") was calculated, and together with the drainage topology, was then used to update the catchment attributes ("Cat_area" and "Cat_a_tot") in the final outlet layer. Eventually, the subset of the preliminary reaches that connect the final lake outlets were extracted to represent the final inter-lake connecting reaches (Fig. 10c).

A few additional attributes were added to the preliminary and final lake outlets with assistance of the existing outlet attributes. Given each outlet, its next upstream lakes were queried based on the values of "D_out_id" of other outlets, and the count of the next upstream lakes was written to the attribute "U_lake_n". The minimum, maximum, and average drainage distances from these upstream lakes ("U_dist_min", "U_dist_max", and "U_dist_avg") were updated using the downstream drainage distance attribute ("D_distance") of the directly connected upstream outlets. The total count of upstream lakes in the entire upstream catchment from the headwater ("U_lak_ntot") and the total count of downstream lakes to the sink or ocean ("D_lak_ntot") were also calculated using the existing outlet topology. The total drainage distance from the outlet to the sink or ocean ("D_dist_sink") was calculated by propagating the drainage distances of each of the downstream reaches ("Rch_length"). Meanwhile, the accumulative lake abundance upstream to each reach ("U_lak_ntot") was also calculated using the topological attributes and assigned to the preliminary and final inter-lake reaches (Fig. 6).

The drainage type for each preliminary and final outlet ("Outlet_type") was next configured based on its hydrological position in the affiliated inter-lake reach network. As explained in Section 2.1 and Fig. 4, the use of inter-lake network eliminated the "isolated" and "inflow-headwater" types, leading to a total of four drainage types for this outlet attribute. The outlet was



assigned to "endorheic" if it coincides with an inland sink and to "coastal" if it touches the MERIT Hydro coastal line. The outlet was assigned to "headwater" if there is no upstream lake but has a downstream reach. All the other outlets were assigned to "flow-through", where they have both upstream lakes and downstream reaches. The spatial distribution of the final outlet types, as well as the composition of their corresponding unit catchment areas, are shown in Figs. 4 and 5, respectively. As illustrated, flow-through lake outlets account for 42% of the total outlets in count, and were drained from nearly 80% of the global lake catchment area.

Due to incomplete digitization, ice cover, and/or seasonal variation, some of the endorheic lakes in HydroLAKES exhibit partial or fragmented water extents. As illustrated by the example in Fig. 11a, many of these incomplete endorheic lake polygons do not contain their associated MERIT drainage sinks. As a result, the outlet derived from the maximum flow accumulation ended up on the incomplete lake boundary, leading to a misassignment of the outlet type as "headwater" or "flow-through" (rather than "endorheic"). Moreover, the catchment delineated based on the misplaced outlet did not enclose the endorheic basin. To tackle this issue, an intermediate outlet type "endorheic-check" was assigned to any outlet that is directly connected to a MERIT sink for visual quality assurance and quality control (QA/QC) in the next step.

**4.4 Quality control for endorheic lakes**

This was the only step in the TopoCat construction that involved manual intervention. In the QA/QC process, each outlet temporarily labeled as "endorheic-check" in the drainage type attribute was visually inspected against high-resolution Google Earth and Esri satellite images. With expert knowledge, this visual inspection aimed at judging whether this label ed outlet is a true outlet of an open lake or an "artifact" pour point of an incomplete endorheic lake (as in the example of Fig. 11a). The outlets verified as the latter scenario were flagged for automated correction. To minimize manual workload and avoid further complications, we performed this QA/QC on the final (rather than preliminary) outlets.

In the following automated correction, each flagged outlet was snapped to its connected sink (the black dot in Fig. 11b). If there are multiple outlets from the same incomplete endorheic lake polygon (which all drain to a single sink), the outlet associated with the largest total catchment area ("Cat_a_tot") was selected for this snapping process. If there are fragmented HydroLAKES polygons representing the same endorheic lake (i.e., with different HydroLAKES IDs), the outlet associated with the largest polygon was selected for snapping. This automated snapping was performed on the preliminary outlets, and the unit catchments were re-delineated based on the updated outlets (drainage sinks in this case). Steps 2 and 3 (Section 4.2 and Section 4.3) were then repeated to accommodate these corrections in all preliminary and final outlet, catchment, and reach attributes. It is worth noting that this QA/QC did not aim at dissolving the fragmented endorheic lake polygons in HydroLAKES. This was to maintain the consistency that every HydroLAKES polygon has at least one outlet and one unit catchment. However, since there was only one outlet snapped to the sink per endorheic lake, only the snapped outlet and its associated unit catchment were corrected to an endorheic drainage type. In other words, for an endorheic lake that has multiple HydroLAKES polygons, only the polygon with a snapped outlet was labeled "endorheic".

Earth System
Science
Data

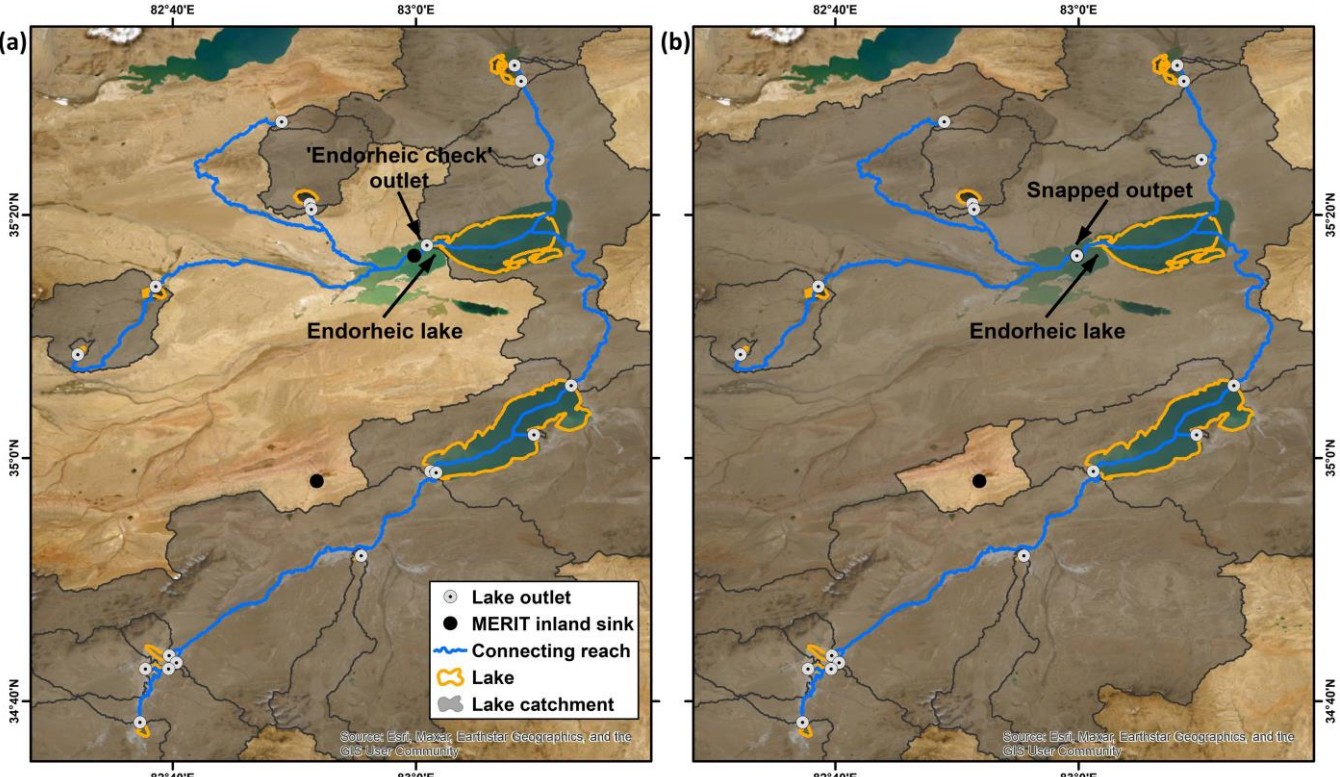

**Figure 11. An example of QA/QC for endorheic lakes.** (a) Derived TopoCat features surrounding an endorheic lake with partial water extent in HydroLAKES. The incomplete polygon for this endorheic lake does not contain its drainage sink (black dot) in MERIT Hydro. As a result, the derived lake outlet before QA/QC (red dot labeled by "endorheic check") is on the incomplete lake boundary and the associated terminal catchment was delineated incompletely. (b) Improvement in the terminal unit catchment and drainage topology after QA/QC, where the "endorheic check" outlet was automatically snapped to the sink and the updated terminal catchment became complete. Here, lake boundaries are the same as HydroLAKES (Messager et al., 2016), and map background from Esri, Maxar, Earthstar Geographics, and the GIS User Community in both figures.

A total of 1,261 outlets were identified as "endorheic-check". Among them, 291 were confirmed to be pour points of incomplete endorheic lake polygons, and 14 of these 291 outlets shared drainage sinks with other outlets. The locations of the other 277 outlets were snapped to their connected drainage sinks, and the drainage catchments were re-delineated. As exemplified in Fig. 11, the result of this QA/QC was a complete endorheic catchment and corrected topological attributes despite an incomplete lake extent in the original HydroLAKES dataset.

## 4.5 Configuring reach types, stream orders, outlet orders, and lake-network basins

The inter-lake reaches in TopoCat depict feasible drainage connections between lakes, whereas the river networks in MERIT Hydro-Vector (Lin et al., 2021) represent, in principle, the actual perennial or intermittent channels (refer back to Section 3.2). Given this logic, we applied MERIT Hydro-Vector to obtain a priori information regarding whether an inter-lake reach in TopoCat is likely a river channel or a non-channelized drainage connection between the lakes. The proportion of each inter-



lake reach that intersects MERIT Hydro-Vector was calculated in percentage of length and saved in the attribute "Rchint_mhv" of the inter-lake reaches.

The Strahler stream order (attribute "Rch_order") was also calculated for each inter-lake reach. The order of each lake outlet ("Out_order") was then assigned as the stream order of the connected upstream reach (whose end node concurs with this outlet). Therefore, the headwater outlets always have a zero outlet order. If multiple reaches converge to the same outlet, the maximum stream order among the confluence reaches was assigned as the outlet order. Finally, the lake-network basins were delineated using the terminal points (inland sinks or coastal outlets) of all inter-lake reach networks. On rare occasions, a

multifurcation lake may infringe more than one reach network; when this occurred, the multiple basins encompassing these reach networks were dissolved into one basin. The "Basin_id" was then assigned to all associated lake outlets, unit catchments, and inter-lake reaches in both the preliminary and final TopoCat.

**4.6 Aggregating attributes at the lake level**

The TopoCat attributes for each lake were configured by aggregating the values of the associated outlet attributes. They include

the counts of directly connected upstream and downstream lakes ("U_lake_n" and "D_lake_n"), the counts of all upstream lakes from the headwater and all downstream lakes to the terminus ("U_lak_ntot" and "D_lak_ntot"), the count of outlets for each lake ("Outlet_n", >1 indicating multifurcation), and the area of the entire lake catchment ("Cat_a_lake", a dissolve of all unit catchments of and upstream to this lake). Lake orders ("Lake_order") were directly obtained from the outlet orders, and in case of a multifurcation lake, its lake order was based on the maximum of its outlet orders. The ID of the inter-basin each

lake belongs to was also assigned according to the basin ID of the associated lake outlet(s).

The drainage type of each lake was configured in the final step. As explained in Section 2.1, we provided two lake drainage types depending on the applied drainage networks: "Lake_type" based on our own inter-lake reaches and "Laktyp_mhv" based on MERIT Hydro-Vector. For most lakes with a single outlet, the value of "Lake_type" was equivalent to the outlet type, but for each multifurcation lake, the value was sorted out by considering the drainage hierarchy among its outlets. As described in

Section 2.2, precedence was given to the outlet type corresponding to the most downstream drainage position. In such cases, "endorheic" and "coastal" outlets have the most weight in determining their lake types, followed by "flow-through" and "headwater" outlets.

The values of "Laktyp_mhv" were modified from "Lake_type" values with reference to MERIT Hydro-Vector (Fig. 2b). MERIT Hydro-Vector contains river networks with variable drainage densities (hereafter "VDD networks"), as well as river

networks with a 25-km$^2$ constant drainage density ("CDD networks"). While the VDD networks are the state-of-the-art global river networks available today, the river topology in VDD networks is occasionally disconnected, and these networks do not cover Greenland and part of other circum-arctic regions. To address the problems, we independently used both VDD and CDD networks to identify lake drainage positions, and then merged the results (i.e., VDD and CDD lake types) to jointly determine "Laketyp_mhv".

Specifically, for any lake where the "Lake_type" value was already the most downstream possible, i.e., "terminal" or "coastal", the same type was assigned to its VDD lake type as well. Lakes that are geometrically off the VDD networks were updated to be "isolated". For the rest of the lakes, the counts of inlets and outlets were obtained through intersecting the lakes with the VDD networks. Note that the count of directly connected upstream lakes in the VDD networks could be different from that based on our inter-lake reach networks. If a lake has at least one directly connected upstream lake, at least one inlet, and also

at least one outlet, the lake was identified as "flow-through" type. The lake was identified as "inflow-headwater", if the lake has at least one inlet and at least one outlet but no directly connected upstream lake. If the lake has only outlets but no directly connected upstream lake or inlet, the lake was identified as "headwater" type. The same process was repeated for the CDD networks to identify the CDD lake types. The final value of "Laktyp_mhv" was then assigned by comparing the VDD and CDD lake types. Similar to the "Lake_type_inter" attribute, we maintained the preference hierarchy towards more downstream

types, where "endorheic" and "coastal" have the most preference, followed by "flow-through", "inflow-headwater", "headwater", and "isolated" types.

## 5 Quality assessment and validation

The quality of TopoCat was first assessed by comparing its lake catchment areas and lake outlet locations with those in HydroLAKES or LakeATLAS with reference to high-accuracy regional hydrography datasets: LakeCat and NHDPlusV2 for

the continental US (CONUS). Since the lake polygons, lake outlets, and catchment or watershed area in LakeATLAS are the same as in HydroLAKES, we only referred to LakeATLAS hereafter, but results are also true for the HydroLAKES. This comparison is followed by a more detailed validation against the lake topology acquired from NHDPlusV2 and LakeCat in a lake-dense area of Minnesota. Finally, the performance of our delineated endorheic basins in TopoCat were assessed against Liu21 endorheic basins 2000 and 2018 for the endorheic Tibetan Plateau (refer to Section 3.2 for validation datasets). Details

of quality assessment and validations are given in the following sections.

### 5.1 Comparison with LakeATLAS

We hypothesize that the overall accuracy of lake outlets and catchment areas in TopoCat, which were derived from the 3-arc-second MERIT Hydro, outperforms those in LakeATLAS derived from the 15-arc-second HydroSHEDS. In addition, the lake catchments in TopoCat were delineated using an iterative process to consider possible multifurcation, so our derived lake

catchment boundaries are likely more accurate and complete. To test this hypothesis, we benchmarked the total lake catchment areas (i.e., the entire drainage areas from the headwater to each lake) derived from LakeATLAS and TopoCat against LakeCat. Following the method performed by Lehner et al. (2022) for validating LakeATLAS, we identified 3991 pairs of lakes in between TopoCat and LakeCat (lake polygons in the latter acquired from NHDPlusV2) which have at least a 90% overlapping area per pair. Values of the lake catchment area attribute in LakeCat were then spatially joined to TopoCat. Since TopoCat

was developed using the same lake polygons as in the LakeATLAS, these lake pairs are exactly the same as those used for





LakeATLAS validation (Lehner et al., 2022), and the only difference is the lake catchment areas used for comparison with those in LakeCat. As shown in Fig. 12a, the agreement of lake catchment areas between TopoCat and LakeCat (Fig. 12a; log-log least-square linear regression R-squared = 0.72, Symmetric Mean Absolute Percentage Error (SMAPE) = 36%, and n = 3991) is overall higher than that between LAKEATLAS and LakeCat (Fig. 12b; log-log least-square linear regression R-

squared = 0.63, SMAPE = 51%, and n = 3991). Fig. 12e compares the cumulative distributions (in relative frequency) of the catchment-to-lake area ratio calculated from TopoCat and LakeATLAS. In TopoCat, the catchment-to-lake area ratios for 80% of the global lakes range from 2 to 70. In LakeATLAS, however, about 5% of the lakes have surface areas larger than their own catchments. This logical error was reduced to less than 0.04% in Topo-Cat, probably due to the use of higher-accuracy hydrography data and a careful consideration of lake multifurcation leading to more complete catchment boundaries.

We also compared the geodesic offsets of LakeATLAS and TopoCat outlets from the benchmark lake outlets. For simplicity, only the primary outlets in TopoCat were used for this comparison in case of multifurcation lakes. For benchmarking, we intersected NHD flowlines with the 3991 NHDPlusV2 lake polygons identified in the previous step. The intersecting point between each lake polygon and the outbound NHD flowline was considered as the benchmark outlet (hereafter "the NHD outlet"). To be consistent with the other datasets, the outlet with the largest upstream catchment area was selected in case of

multiple NHD outlets for a single lake. A total of 2569 NHD outlets were generated across CONUS, and the average offset to LakeATLAS and TopoCat outlets is 577 (±1402) m and 393 (±1395) m, respectively, indicating an overall better accuracy of lake outlet locations in TopoCat. Figure 12c shows a few examples in northern Wisconsin, where the TopoCat outlets are in much closer proximity to the NHD outlets than the LakeATLAS outlets. In ~76% of the 2569 cases, the offsets between TopoCat and NHD outlets are ~361 (±735) m smaller than the offsets between LakeATLAS and NHD outlets (Fig. 12d),

confirming that the overall quality of the outlet locations in TopoCat is superior to that provided in LakeATLAS.







**Figure 12. Improvements in TopoCat catchment areas, outlet locations, and catchment-to-lake area ratio in comparison with the information in LakeATLAS.** (a) Comparison of total lake catchment areas in between TopoCat and LakeCat (benchmark). Only lake polygons from both datasets overlapping at least 90% of the area were considered for this comparison (n = 3991). (b) Same as the Fig. 12a except the comparison was made between LakeATLAS and LakeCat. (c) Locations of lake outlets in TopoCat and LakeATLAS, with respect to the outlets derived from NHD (benchmark). Here, NHD outlets were located using the intersecting points between outbound NHD flowlines and the NHD waterbodies showing at least 90% area overlap with lake polygons in LakeATLAS. Outlets of 2596 NHD waterbodies were identified from 3991 overlapped lakes (lake boundaries are the same as HydroLAKES; Messager et al., 2016, and map background from Esri, Maxar, Earthstar Geographics, and the GIS User Community). (d) Offsets (in geodesic distance) of LakeATLAS vs TopoCat outlets from NHD outlets. Points above the diagonal line indicate that TopoCat outlets are closer to NHD outlets than LakeATLAS outlets are. (e) Comparison between the cumulative distributions of the catchment-area to area-area ratio calculated from TopoCat and LakeALTAS. About 5% of the lakes in LakeATLAS have larger lake areas than their catchment areas, whereas in TopoCat such unreasonable cases are less than 0.04%.



## 5.2 Validation of lake topology

Validating lake topology is challenging because there are only a few regional hydrography datasets that consider lake components, and additionally, lake topology is not readily available in these datasets and retrieving it requires a substantial amount of pre-processing. In this study, we benchmarked our TopoCat lake topology against the reference data in a lake-dense area of Minnesota (Fig. 13), located at the intersection of the Canadian Shield (the Laurentian Plateau) and the Prairie Pothole Region. We selected this validation site for two reasons. The first reason was the availability of high-quality hydrography data

(in this case, NHDPlusV2 and LakeCat). The second reason is that the accuracy of TopoCat is expected to be relatively low in this region given the difficulty of using gridded hydrography to portray the geological and hydrological complexity in such lake-rich areas. Thus, our rationale is that if the performance of TopoCat is acceptable in this lake-rich area, it may work equally well or better for the rest of the world.

The TopoCat and benchmark datasets were pre-processed to enable a comparison at equivalent levels. First, the waterbodies

in NHDPlusV2 were intersected with the lake polygons in TopoCat within the validation site to retrieve the common lakes in both datasets. A linking ID between the common lakes was established. In occasional cases where one TopoCat lake covers multiple NHDPlusV2 lakes, the link was established for the most downstream NHDPlusV2 lake among the ones intersecting TopoCat. This way, the topological orders among the linked subset of NHDPlusV2 lakes were still maintained. A total of 427 linked lake pairs were established between NHDPlusV2 and TopoCat, where 112 intersect the NHDPlusV2 flowline network

(hereafter referred to as "on-network lakes"). Topological orders of the on-network lakes were obtained from the NHDPlusV2 dataset using the topological orders of their associated catchments. Several on-network NHDPlusV2 lakes intersect no lake in TopoCat. In such cases, we assumed that the lake is absent in TopoCat (i.e., the HydroLAKES polygons), so such lakes were removed from the assessment, and the topological orders of the remaining on-network lakes were updated accordingly.

The topological orders of the off-network lakes in NHDPlusV2 (315 in total) were obtained from the LakeCat database (i.e.,

from associated lake catchment orders). Similar to the pre-processing for on-network lakes, any off-network lake intersecting no lake in TopoCat was removed from the assessment, followed by an update of the topological order. The updated off-network lake topology was then appended to the updated on-network ones to form a completely pre-processed validation or benchmark lake topology (hereafter "NHDPlusV2 topology"). It is noteworthy to mention that, only on-network waterbodies were burnt into the DEM in development of the NHDPlusV2 hydrography dataset (i.e., flow direction, flow accumulation). On the other

hand, the LakeCat directly used the flow direction of the NHDPlusV2 to derive the topology of off-network lakes, instead of developing a hydrologically adjusted flow direction through burning off-network lakes in DEM. Therefore, the reference topology of the on-network lakes is more reliable than the off-network lakes. Akin to the benchmark data processing, TopoCat lakes intersecting no lakes in NHDPlusV2 were excluded from the assessment, and all attributes in TopoCat were updated to accommodate these changes.

The updated TopoCat topology was compared with NHDPlusV2 to assess the skill in identifying the downstream lake, as well as the skill in identifying the upstream lake abundance (i.e., the total count of upstream lakes from the headwater to each lake).

Earth System
Science
Data

The spatial distributions of these skills are shown in Fig. 13a. Here, thick lines represent correct connections made by TopoCat and thin lines for incorrect connections. About 80% of the on-network lakes in TopoCat are correctly connected to the next downstream lake, and for off-network lakes the skill is ~55% (Fig. 13b). Different colors of lakes show the skill in capturing

the upstream lake abundance (Fig. 13a). We considered TopoCat to be skillful when the difference in the total upstream lake count between TopoCat and the benchmark data is within ±1. Given this tolerance, about 76% of the lakes in TopoCat capture the total count of upstream lakes, which cover ~88% of the lake surface area within this region (Fig. 13c).

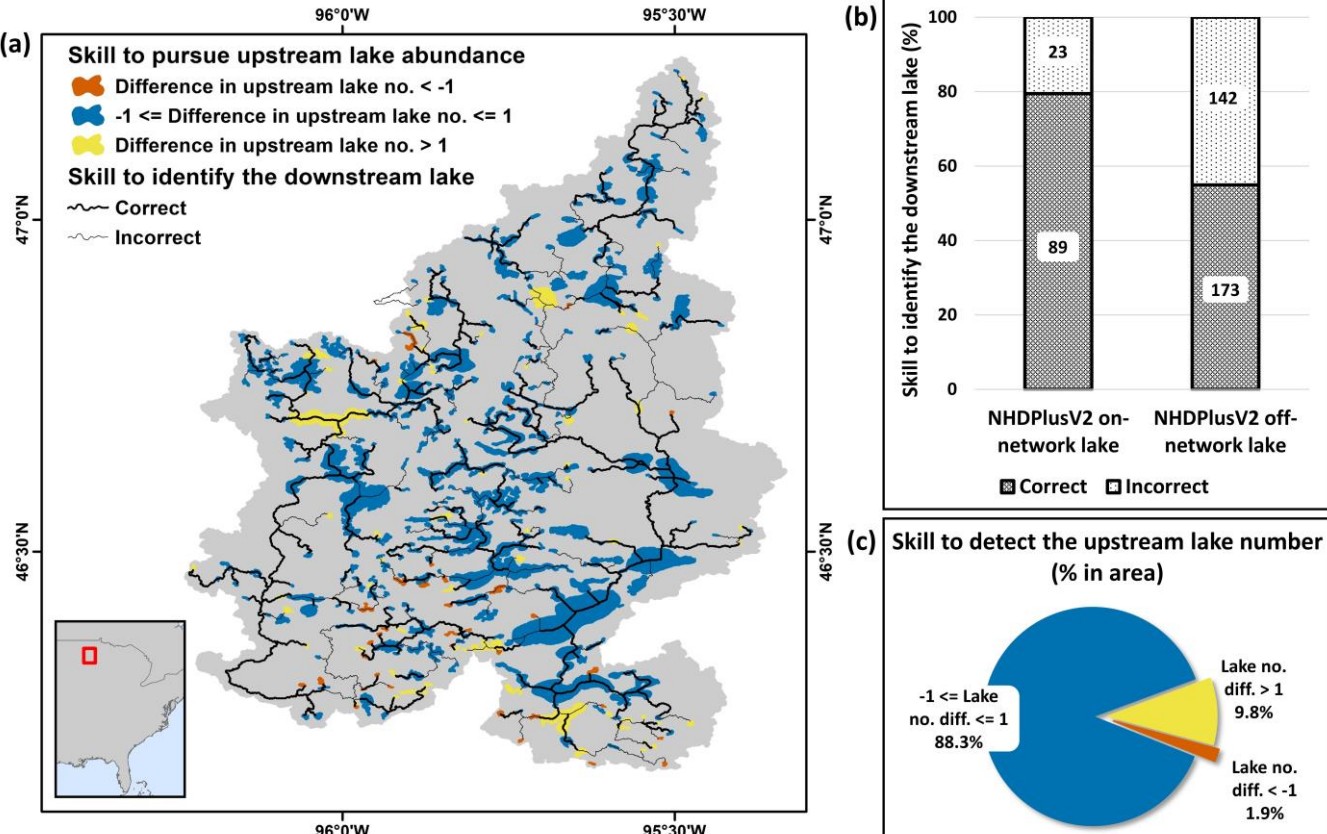

**Figure 13. Skill of TopoCat lake topology in the lake-rich Prairie Pothole Region.** (a) Comparison with the benchmark lake topology
derived from NHDPlusV2 and LakeCat (lake boundaries are the same as HydroLAKES; Messager et al., 2016). Lines illustrate the accuracy of detecting the next downstream lake, whereas lake colors indicate the accuracy of detecting the total count of upstream lakes. (b) Skill of detecting the next downstream lake. The total number of lakes that correctly (and incorrectly) identified the downstream lakes shown on the bars. (c) Distribution of lake area according to the accuracy of detecting the total upstream lake count.

**5.3 Validation of endorheic basins**

Since the Tibetan Plateau is one of the most endorheic lake-rich regions in the world, we selected this region to validate the endorheic basins derived in TopoCat (as in the lake-network basin layer), using the benchmark data "Liu21 endorheic basins 2000 and 2018" (Liu et al., 2021; refer back to Section 3.2). As a preparation of catchment delineation, the raw DEM is usually processed to fill any trivial depressions or artifact pits before generating hydrologically realistic flow direction and flow



accumulation. However, during this pit-filling process, true inland depressions may also be removed from the DEM (Grimaldi et al., 2007), which makes it challenging to delineate endorheic catchments properly. Liu et al. (2021) used a lake-oriented approach to delineate endorheic basins on the Tibetan Plateau. In their method, individual terminal lakes were first mapped from spectral images, and their associated endorheic catchments were then delineated accordingly. A total of 253 endorheic basins were selected from TopoCat on the studied Tibetan Plateau. Unlike Liu et al. (2021), the topography modification in MERIT Hydro was not tailored for each observed endorheic lake. Instead, Yamazaki et al. (2019) assumed that a significant volume change in topography was required to breach a true inland depression to its adjacent basin. After trial-and-error tests, MERIT Hydro adopted a threshold modification volume (2,500,000 $m^3$) to automatically separate true depressions from the dummy ones. Although this automatic approach was proven to be effective at a global scale, it occasionally failed to detect true depressions on the Tibetan Plateau, ultimately connecting some of the actual endorheic basin to its neighboring catchments. This type of error particularly occurs for small endorheic basins, where an effective pit-fill volume is smaller than the threshold modification volume.

The boundaries of endorheic basins in TopoCat are visually compared with those of Liu21 endorheic basins 2018 (the validation benchmark) in Fig. 14a. The numbers of basins in TopoCat and the benchmark data on the endorheic Tibetan Plateau are 259 and 245, respectively, and in 13 cases, the latter covers multiple basins in TopoCat. In such cases, we merged the TopoCat basins enclosed by the corresponding benchmark basin to make the comparison feasible. The filling colors in Fig. 14a illustrate the Critical Success Index (CSI) value for each TopoCat endorheic basin, here calculated as:

$$CSI = \frac{H}{H+M+F} \qquad (1)$$

where H is the area overlapped by TopoCat and the benchmark basins, M is the benchmark basin area not overlapped by the corresponding TopoCat basin, and H is the TopoCat basin area not overlapped by the corresponding benchmark basin. Although the input hydrography and lake mask data are different in the two datasets, their basin boundaries show a fairly good agreement. The overall CSI value of all the 245 endorheic basins is ~0.85, with the individual basin CSI values ranging from 0.03 to 0.99. The largest difference occurs in the endorheic basin of the Pangong lake, which was misidentified as part of the exorheic Indus Basin in TopoCat. Our visual examination attributed this omission error to the automated process of dummy depression removal in MERIT Hydro, where the adopted threshold modification volume was too large for this basin.

Figure 14b further shows the scatter-plot comparison between the TopoCat and benchmark endorheic basins. Here TopoCat basins were validated against Liu21 endorheic basins for both year 2000 (log-log least-square regression R-squared = 0.86; SMAPE = 17.5%; n = 246) and year 2018 (log-log least-square regression R-squared = 0.87; SMAPE = 16%; n = 245). During these two decades, lake expansions on the inner Tibetan Plateau triggered several cases of lake capture and basin coalition (Liu et al., 2021). These ongoing basin re-organizations explain some of the basin area discrepancies. Despite such uncertainty, our delineated endorheic basin areas in TopoCat agree well with the benchmark basin areas in both 2000 and 2018 except a few outliers such as Pangong lake. TopoCat shows slightly overestimated endorheic basin areas in a few cases. In these cases,





small endorheic basins were merged with the adjacent basins during the dummy depression removal process, thus increasing the sizes of the adjacent endorheic basins.

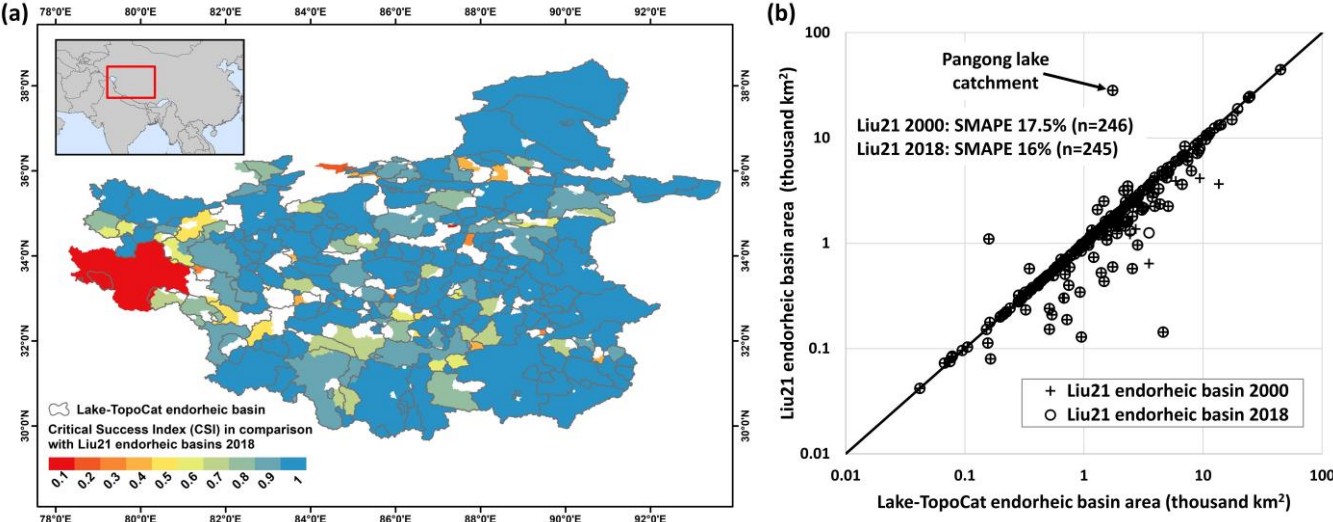

**Figure 14. Skill of TopoCat endorheic basin delineation on the Tibetan Plateau.** (a) Comparison of Lake-TopCat endorheic basin boundaries with the benchmark data (Liu21 endorheic basins 2018). Different colors illustrate the Critical Success Index (CSI) values, calculated based on the area of the benchmark basin overlapped by the corresponding TopoCat basin (see Eq. (1)). (b) Cross-comparison between TopoCat basin areas versus those in Liu21 endorheic basins 2000 and 2018, respectively.

## 6 Applications and limitations

We developed Lake-TopoCat v1.0, which contains the topological orders, drainage paths, and unit catchments for 1.4 million lakes and reservoirs larger than 10 ha in the world (HydroLAKES v1.0). Among them, the inter-lake reaches are one of the key features. About 3 million inter-lake reaches were delineated, ~80% of which are shorter than 10 km. These reaches stretch ~10 million km, which is about 4.6 times longer than global drainage networks with rivers wider than 30 m (Altenau et al., 2021; Allen and Pavelsky, 2018). The detailed inter-lake drainage networks in TopoCat allow us to decipher lake drainage topology, which may help model how aquatic species possibly migrate upstream or downstream through the connecting reaches as a response to climatic or ecological changes. This lake topology is required to integrate medium- and small-sized lakes and reservoirs, which are largely lacking in the existing routing models, into the global river networks. Such integrated lake-river networks will be essential to route water storage variability in lakes and reservoirs, such as observed by satellite altimeters including the Surface Water and Ocean Topography (SWOT) mission (Biancamaria et al., 2016), into hydrological models and thus benefit the estimates of river discharge, fluvial sediment transport, and carbon flux cycling. For instance, Abril and Borges (2018) studied carbon leaks from flooded land, and argued that a global topology of fluvial and lacustrine wetlands is necessary to improve the quantification of hydrological carbon fluxes in flooded ecosystems. This argument also applies to lakes and reservoirs, which are important for carbon sequestration.



The spatiotemporal variability of lake-river connections is important to understand surface water dynamics, especially in the Arctic and arid regions, where lake-river connections are intermittent and complex. In Arctic environments, the fluvial and glacio-fluvial processes heavily depend on lake-river connectivity. For example, Dolan et al. (2021) found that about 10% of the lakes in Colville Delta in Alaska have variable temporal connectivity, and the lake-river connectivity is highly correlated with river discharge. This connectivity influences ice breakup timing, where ice in highly connected lakes breaks up earlier than in less connected lakes. The inter-lake drainage networks of TopoCat offer all possible hydrologic connections among global lakes, which can serve as a priori information to monitor the intermittency and dynamics in fluvial-lacustrine connectivity. The lake topology is essential to identify upstream lake abundance as well (Gardner et al., 2019), which is useful in understanding the density of lakes in a region and their accumulative downstream impacts. From TopoCat we found that ~82% of the global lakes have a drainage proximity to other lakes of less than 10 km, and the percentage increases to ~95% if the proximity increases to 100 km. Lake-TopoCat data can be used to analyze geomorphic scaling principles that have been shown to be related to lake formation processes and even lake water residence time (Seekell et al., 2021).

Our delineated lake catchments reveal that nearly 57% of the Earth's landmass (excluding Antarctica) drains into a lake larger than 10 ha. These unit lake catchments will allow us to understand lake water mass balance and water quality changes by explicitly considering the geophysical and anthropogenic characteristics of the lake drainage basins and thus the contributions of the basins. The amount of cropland in a lake catchment can affect the amount of nutrient-driven primary production in lentic ecosystems (Balmer and Downing, 2011). Similarly, the amount of wetland coverage in a lake's catchment controls the amount of dissolved organic material transported downstream, cascading to primarily production in lakes, and ultimately carbon dioxide emissions from these waterbodies (Borges et al. 2022; Maberly et al., 2013). The geomorphology (e.g., gradient, altitude) of a lake's catchment may also play a role in controlling lake biogeochemistry and greenhouse gas emissions of lakes (Casas-Ruiz et al., 2020; Lapierre and Giorgio, 2012).

A few recent studies (e.g., GLCP and LakeATLAS) used the lake drainage basins to compute basin-averaged river area, temperature, precipitation, population, and many other hydrological-limnological variables (e.g., Lehner et al., 2022; Meyer et al., 2020). However, lake catchment perimeters are not directly available from these studies and often do not accurately represent the actual drainage areas of the lakes (Meyer et al., 2020). Together with other thematic datasets, the high-resolution (i.e., 3-arc second) lake catchment boundaries in TopoCat provide users with more accuracy and flexibility to customize the calculations of basin-level limnological and socioeconomic attributes. Furthermore, the lake drainage topology along with lake basin boundaries will help users better understand lake water quality issues, such as tracking temperature and nutrient flows from the upstream lakes and catchments that affect downstream lake algae contributions and water color (e.g., Hou et al., 2022; Yang et al., 2022).

Despite the improvements over existing datasets identified above and a variety of potential applications of TopoCat, we would like to acknowledge a few notable limitations. First, TopoCat aims to construct a priori drainage topology and connectivity among global lakes. Thus, the "headwater" extent in our delineated inter-lake reach network was determined by the presence of the most upstream lakes rather than river channels. This is partially due to the ambiguity of what and where should be



considered a headwater river and the variability of drainage density across the continents (Lin et al., 2021). However, our lake-based drainage network can be potentially nested into a broader and more detailed drainage network to enable a more complete global lake-river integration.

Second, although our algorithm includes a mechanism to consider possible lake multifurcation, and this mechanism design did successfully capture some of the multifurcation cases (see the example in Fig. 3), we acknowledge that our lake multifurcation attribute has not been fully validated. Our identified multifurcation was based on MERIT Hydro. Although MERIT Hydro incorporated long-term maximum water extents to ensure drainage connectivity and to calibrate drainage directions, topographic errors still remain, particularly in floodplains and wetlands with large drainage complexity and uncertainties.

Some of the major multifurcation lakes, such as Lakes Vesijako (61.38° N, 25° E), Lummene (61.48° N, 25.05° E), and Diefenbakern (51.03° N, 106.84° W), were unfortunately not captured by our automated algorithm. These errors may be due in part to errors in lake mask polygons from HydroLAKES. HydroLAKES does not always represent precise lake boundaries, and some polygons span multiple lakes and even infringe on other rivers and drainage watersheds (leading to artifacts or pseudo bifurcation issues). These errors have not been fully considered in the current TopoCat version and may require future

manual QA/QC and validation. A higher-resolution, remote-sensing-based, and quality-controlled lake mask, such as the SWOT a priori lake database (Sheng et al., 2016), may also help improve the representation of lake multifurcation.

Third, the accuracy of the delineated basin boundaries in Lake-TopoCat v1.0 depends on the quality of MERIT-Hydro, which applied an automatic approach to remove dummy depressions from the DEM (Yamazaki et al., 2019). This automated DEM processing occasionally lumped actual small endorheic basins to their adjacent exorheic or endorheic basins, which could lead

to incomplete lake boundaries and misclassification of the lake drainage types. For instance, the Pangong Lake (33.72° N, 78.9° E) on the Tibetan Plateau is an endorheic lake, but in MERIT Hydro the Pangong basin was incorrectly merged to the Indus Basin.

Last, the current version of Lake-TopoCat (v1.0) emphasizes lake drainage topology and catchments, rather than explicit lake-river integration and segmentation. Therefore, only lake outlets were considered at this moment. While the position of the lake

inlet is subject to upstream and downstream shifts as the lake area expands and shrinks, the outlet position is often more stable, particularly in reservoirs and perennial lakes. Located at the most downstream boundary of a lake, the outlet also determines the full catchment domain governing the water mass and quality of the lake, and it is more straightforward to use outlets to configure lake drainage topology and derive unit lake catchments. For these reasons, we consider the delineation of lake outlets to be both more meaningful and practical. However, we admit that the addition of lake inlets is necessary for a more complete

segmentation and coupling between lentic and lotic units along the drainage networks, and therefore, the inclusion of both lake inlets and outlets will be considered in our future lake-river harmonization efforts.

## 7 Database availability

The Lake-TopoCat v1.0 database and its possible future versions are available in both shapefile and geodatabase formats through Zenodo (https://doi.org/10.5281/zenodo.7420810; Sikder et al., 2022) under the Creative Commons Attribution 4.0

International license. While the method to develop TopoCat is generic and adaptive, the version of TopoCat (v1.0) illustrated in this paper was generated for the global lake mask HydroLAKES v1.0 using primarily MERIT Hydro v1.0.1. As our intention is to make TopoCat an input-data-flexible and recursively improved database, future versions may be produced using other global or regional lake masks and hydrography datasets. To avoid version ambiguity, the specific version illustrated in this paper should be referred to as "HydroLAKES-TopoCat v1.0", and "Lake-TopoCat" should be used to present the generic or

collective version of our current and future lake topology and catchment data products.

## 8 Author contribution

MSS contributed to data curation, formal analysis, investigation, methodology, programming, quality assurance, quality control, validation, visualization, writing – original draft preparation, and writing – review and editing. JW contributed to conceptualization, data curation, formal analysis, funding acquisition, investigation, methodology, programming, project

administration, quality assurance, quality control, supervision, validation, visualization, writing – original draft preparation, and writing – review and editing. GA contributed to data curation, methodology, validation, and writing – review and editing. YS contributed to data curation, methodology, and writing – review and editing. DY contributed to data curation, methodology, and writing – review and editing. CS contributed to data curation, methodology, validation, and writing – review and editing. MD contributed to quality control, and writing – review and editing. JFC contributed to methodology, and writing – review

and editing. TMP contributed to methodology, and writing – review and editing.

## 9 Competing interests

The authors declare no conflict of interest.

## 10 Disclaimer

Authors of this paper claim no responsibility or liability for any consequences related to the use, citation, or dissemination of

Lake-TopoCat v1.0.



## 11 Acknowledgements

This work was supported by NASA Surface Water and Ocean Topography (SWOT) Grant (#80NSSC20K1143). The authors thank Kai Liu at Nanjing Institute of Geography and Limnology, Chinese Academy of Sciences for providing the validation data for endorheic basin boundaries on the Tibetan Plateau. The authors are also grateful to Bernhard Lehner at McGill University, Michael F. Meyer at USGS, R. Iestyn Woolway at Bangor University, Daisuke Tokuda at The University of Tokyo, and Alberto V. Borges at University of Liège for their constructive insights on the construction and applications of Lake-TopoCat v1.0.

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
