# Peer review of "Lake-TopoCat: A global lake drainage topology and catchment database"

_Earth System Science Data, 2022_

## Author Response (AR1)

We would like to thank the editor and the two anonymous reviewers for their careful evaluation of this manuscript, and for constructive feedbacks. We addressed the issues as per the suggestions of the reviewers, after thoroughly reading each comment. The changes in the manuscript include:

1) A few sub-sections in the updated manuscript have been rearranged and merged with the other sections. For example, sections 3.1 and 3.2 of the original manuscript have been merged with the methodology (i.e., previously section 4, and now section 3 in the updated manuscript).

2) In addition to that, Figures 4, 11, and 13 have been updated/corrected.

3) To respond to reviewer 2's comment, we have introduced a new attribute to the lake polygon named 'Lperm_glcp'. This new attribute quantifies the lake dynamics in terms of the average proportion of the lake area being permanent water, derived from the global lake area, climate, and population dataset (GLCP; Meyer et al., 2020, doi:10.1038/s41597-020-0517-4).

The resubmission of this work contains a 'tracked-changes' version and a 'cleaned' version of the updated manuscript. Further details of the modifications have been provided in the response below to each comment of the reviewers with 'green' colored text.

**Reviewer#1**

Summary Comment – the dataset, Lake-TopoCat will be a valuable contribution to support integrating lakes into hydrological modeling efforts at large scales. The paper is generally well-written. I have some clarifying questions/comments

Response: Thank you so much for your compliment about this study. We sincerely appreciate your time and valuable feedback. We have gone through each of your comments and addressed them accordingly. Please see our responses to your comments.

Paper Organization – this was the paper's biggest issue, I found that as currently organized a lot of the information ended up being repeated multiple times, while I was often left with questions because the full method description was often spread over multiple sections. I kept writing questions only then to have them answered 10 pages later. I have a few suggestions below which I think would help quite a bit.

Response: We agree that section rearrangement makes the manuscripts more readable. We appreciate your kind suggestions regarding the presentation of the paper. We have implemented all the suggestions, except the comments regarding section 2. Please see our responses below to your comments regarding the organization of the paper.

Section 2 – I found section 2 quite confusing to read with a lot of unanswered questions that were then answered in sections 3 and 4. I recommend switching the order so explaining how the datasets were developed before the authors present the final datasets, so sections would be: 1, 3, 4, 2, 5. I think the maps and pie charts will be much more effective if readers understand how they were derived first.

Response: Thank you for your suggestion, but we purposely placed 'Data description and structure' as section 2 in the manuscript before the full methodology section, assuming that not all users of this database are interested in, or need to be familiar with, the technical details of the methodology. We believe that the 'Data description and structure' section will provide a more efficient and wholistic overview of the database to those users. Accordingly, key definitions are introduced in section 2 as the dataset is being demonstrated, and critical methods are also explained in section 2 but only at the conceptual level and only when it is necessary for the understanding of the data logic and usage. Due to the complexity of this dataset, we prefer this style after careful deliberation.

Overall, we believe this structure fulfills the following two purposes: (1) users who are only interested in the usage of this dataset should be able to obtain a good comprehension of the data structure and functionality by only reading section 2; and (2) users who are more interested in how each function was enabled can continue reading the technical details elaborated in the next method section. We hope this makes sense to the reviewer. For improved clarity, we have explained the reason for this placement in the introduction section:

"*For user convenience, the following sections first provide an overview of the components, structure, and functionality of Lake-TopoCat (section 2), before elaborating detailed input data sources and methodology (section 3). They are then followed by quality assessment and technical validation (section 4). We conclude with a discussion on Lake-TopoCat's potential applications and limitations (section 5).*" (Line 120-123)

Sections 3 and 4 – I would recommend combining sections 3.1 and 3.2 and 4. Having the data inputs with some methods description, separated from section 4, which then further described methods, just created more confusion.

Response: We agree with this suggestion. Sections 3.1 and 3.2 have been merged with section 4. In the updated manuscript, section 3.1 and 3.2 are now under 'Input data sources' (i.e., section 3.1 in the updated manuscript), which is under 'Methods' (i.e., section 3 in the updated manuscript). The new section numbers of section 3.1 and 3.2 are 3.1.1 and 3.1.2, respectively. Section 4 (i.e., Methodology and algorithm) of the original manuscript is now section 3.2 (i.e., Dataset development and algorithm) in the update manuscript.

Section 3.3. – This section raised more questions than it answered. I would recommend moving section 3.3 to section 5, which would put all of the validation description in the same place.

Response: As recommended, we have now merged section 3.3 with section 5 in the updated manuscript. Section 5 became section 4 in the updated manuscript, and section 3.3 assigned as section 4.1 (i.e., Validation data) under section 4 (i.e., Quality assessment and validation) of the updated manuscript.

Minor Comments

Line 17 – change "communicate" to "interact" – the term communicate seems a bit odd here.

Response: Thank you and we have changed "communicate" to "interact".

Line 34 – the definition of lakes needs to be expanded here. How are the waterbodies referenced distinguished from wetlands, for example, which are barely mentioned.

Response: Thank you for this important question, although distinguishing lakes from wetlands is beyond the goal of this dataset. Definitions of lakes, ponds, and wetlands found in literature are often lacking consensus (i.e., Richardson et al., 2022, RAMSAR convention, etc.). Since the current version of TopoCat is developed on HydroLAKES, our studied lakes comply with the definition and protocol of HydroLAKES. Here we quote:

"*In terms of distinguishing lakes from wetlands, it is not the goal of HydroLAKES itself to determine a process-based separation between these features. Rather, HydroLAKES relies on the given distinction provided in the utilized source datasets, all of which contained an explicit "lake" category that by design did not include wetlands.*" (HydroLAKES technical documentation version 1.0).

For this reason, our referenced water bodies are natural lakes, ponds, and manmade reservoirs, which are not intended to include wetlands.

Line 52 – what does the term "quality-assured" mean here? Please clarify in the text or remove the term.

Response: The term "quality-assured" has been removed from the abstract.

Line 72 – Change km2 to ha to match the prior dataset descriptions

Response: This unit of the area has been updated from square kilometers to ha.

Figure 4 – add a pie chart of the % distribution, this figure is particularly important since the percents will be count instead of area based.

Response: As suggested, a pie chart has been added to Figure 4 to show the counts of different types of outlets.

Figure 6 – Is this showing the line shapefile? The color stretch is a little challenging to see, what are the black areas? Consider changing the black areas to gray like in Figure 7.

Response: Yes, Figure 6 shows the inter-lake reaches as stated in the figure caption. Thanks for the suggestion. We have tried using a gray background color to be consistent with Figure 2, 4, 5, and 7. Please see Figure R1 below. While a gray color background appears effective in showing the ordinal lake and outlet types (as in the other figures), it is not as visually satisfying for illustrating the gradual increase of accumulative lake abundance (akin to stream order) along the reach networks. In comparison, we feel the black ground (Figure R2) works slightly better for this purpose. This is because a darker hue in the background tends to suppress the appearance of lower stream-order reaches (e.g., 0-1 in navy or dark blue), which ends up enhancing the appearance of higher stream-order reaches (e.g., 2-3, 3-4, etc. in bright yellow to red). This way, the gradual increase of stream order is shown with a more aesthetically appealing and clearer transition (at least to us). We also experimented with changing the color stretch but felt challenging to find a better scheme than the one already used in our previous figure. If the reviewer agrees, we would like to keep the original figure 6 as it is. But to avoid confusion, we have added a sentence in the caption of Figure 6 saying "*Land background is displayed in black to enhance the visual transition from lower to higher upstream lake abundance (akin to increasing stream order).*"

[Figure]

Figure R1. Global map of inter-lake reaches in TopoCat. Reach colors illustrate the accumulative lake abundance or (count) upstream to each inter-lake reach. The histogram on the lower-left corner shows the frequency of upstream lake abundance. A gray land background is applied.

[Figure]

Figure R2. Global map of inter-lake reaches in TopoCat. Reach colors illustrate the accumulative lake abundance or (count) upstream to each inter-lake reach. Land background is displayed in black to enhance the visual transition from lower to higher upstream lake abundance (akin to increasing stream order). The histogram on the lower-left corner shows the frequency of upstream lake abundance.

322-23 – What does "quality-controlled" mean here? Please clarify in the text or remove the term.

Response: The term "quality-controlled" has been removed from the text.

Figure 11b – There is a spelling mistake, change to "Snapped outlet"

Response: Thank you for noticing the typo. We have corrected the spelling mistake in the updated Figure 11b.

Section 5 – how much do you think the DEM quality is impacting the outlet ID accuracy?

Response: To show the impact of the DEM quality on delineated catchment and outlet locations, we compared the Lake-TopoCat with LakeATLAS. The lake outlets in Lake-TopoCat and LakeATLAS were derived from 3 arc-second MERIT Hydro and 15 arc-second HydroSHEDS,

respectively. Besides the spatial resolution, the MERIT Hydro eliminated major error components from input DEM, including absolute bias, stripe noise, speckle noise, and tree canopy height, which often distort surface drainage directions and fragment drainage paths. To compare the accuracy of outlet locations derived from MERIT Hydro and HydroSHEDS, we calculated the geodesic distance of each outlet from the reference outlet. Here, the reference lake outlets were obtained from the intersecting points of the NHD waterbodies with NHD outbound flowline from waterbodies. The distance between outlets from MERIT Hydro (i.e., TopoCat) and NHD is significantly shorter than the distance between outlets from HydroSHEDS (i.e., LakeALTAS) an NHD (Figure 12c, 12d), indicating that the DEM quality has a positive impact on the accuracy of the outlet locations. We found that in ~76% cases the MERIT Hydro improved the location of the derived lake outlet (i.e., outlet location closer to the reference) than the HydroSHEDS while comparing with the reference (in Line 686-688). We also acknowledge that despite these improvements, MERIT DEM as a global product is not always comparable to the qualities of local high-resolution DEMs, which is reflected by the scale of the remaining errors benchmarked against the NHD derivative (section 4.2).

Figure 12 – add the R2 and SMAPE like in a and b to panel d

Response: The panel (a) in Figure 12 shows comparison of the lake catchment areas derived from the Lake-TopoCat with the reference (i.e., NHD). Similarly, panel (b) shows the comparison between lake catchment areas derived from the LakeATLAS and the same reference. In these two panels we provided the R2 and SMAPE to show how much the lake catchment areas derived from these two datasets are off from the reference lake catchment area.

On the other hand, the panel (d) in Figure 12 shows the comparison between how off the Lake-TopoCat outlet location from the reference (i.e., NHD) with how off the LakeATLAS outlet location from the same reference. In this panel the accuracy of the Lake-TopoCat outlet locations has been directly compared with the accuracy of the LakeATLAS outlet locations. A higher R2 value and a lower SMAPE value in this panel would mean that there is no difference in the accuracy of the outlet locations between TopoCat-NHD and LakeATLAS-NHD. It is clearly visible that in most cases the offsets between LakeATLAS and NHD outlets are larger than the offsets between TopoCat and NHD outlets. Therefore, calculating R2 and SAMPE is not very relevant for this panel (it may instead add more confusions to the point we are conveying).

Figure 13b – change the bar graph labels. I understand the current labels represent the source, but at first glance it looks like the accuracy of the NHDPlus versions is what is being presented.

Response: Thank you so much for the suggestion. We agree that the labels on the bar chart are misleading. Both labels have been changed in the updated Figure 13b.

**Reviewer#2**

The article provided a comprehensive dataset of lake drainage topology which receives relatively less attention but will largely help with routing models and environmental studies. The study made use of many recent lake and river datasets, and supplemented them with important lake topology attributes. The workflow is clearly described, the QA/QC processes are complete, and the paper is easy to follow although it is a technical paper. In general, the dataset is promising and quite contributable to the hydrological modelling society. Some issues remain in the current version of the article and I suggest the article be considered publication after minor revision.

Response: Thank you very much for your inspiring words about this research and your comments. We carefully read your comments and addressed them accordingly. Please see our responses to your comments below.

1.  It seems this dataset is most tailored to routing modellers. However, it is not explicitly mentioned how the outlet/inlets locations can be useful. In addition, what about other potential application areas? Can the authors provide a few sentences in the end of the Abstract (or discussions in the main text) to guide more general uses of Lake-TopoCat?

    Response: Thank you for this suggestion. We've added the following sentences at the end of the abstract to suggest a few relevant applications of Lake-TopoCat:

    "*With such unprecedented lake hydrography details, Lake-TopoCat contributes towards a globally coupled lake-river routing model. It may also facilitate a variety of limnological applications such as attributing water quality from lake to basin scales, tracing inter-lake fish migration due to changing climate, monitoring fluvial-lacustrine connectivity, and improving estimates of terrestrial carbon fluxes.*" (Line 29-33)

2.  The structure and organization of the article are a bit confusing. For instance, in Lines 326-327, the sentence is a general introduction of the types of data, instead of a summary of the 3.1 section. The data description part contains too much analysis and classification results, which are hardly comprehensible without referring to Sections 3 and 4. The quality assessment and validation results are mixed with the experimental design and data processing, blurring both the purposes and the results. The data and methodology parts are not clearly separated in the two sections, causing inconvenience in looking for key messages and understanding the design of the workflow.

    Response: Thank you for this comment. We very much agree rearranging some of our sections will improve the clarity and readability. Accordingly (also in response to Reviewer 1's comments), we have absorbed the original input data section into the method section and the validation section. Specifically, the sections of input data sources (i.e., sections 3.1 and 3.2 of the original manuscript) are now part of the methodology section, and the validation

data (i.e., section 3.3 of the original manuscript) is now under the quality assessment and validation section. This way, the methodology and the validation are more clearly separated.

3.  The dataset is using the original HydroLAKES lake IDs to link to the features and attributes of HydroLAKES. It would be better if the authors could insert an additional table of existing HydroLAKES attributes so that the comprehensiveness of the dataset can be better evaluated.

    Response: The original HydroLAKES lake IDs were preserved in the Lake-TopoCat attribute so that the user can easily join TopoCat with HydroLAKES and jointly use their attributes. We believe that adding HydroLAKES attributes to Lake-TopoCat is not necessary, and it will just create a duplicate repository of HydroLAKES (which is something we try to avoid). Moreover, other datasets based on HydroLAKES such as GLCP (Meyer et al., 2020) and GLEV (Zhao et al., 2022) also do not contain any HydroLAKES attribute except the lake IDs. We hope the reviewer finds our rationale acceptable.

4.  It is necessary to mention whether the individual lakes are permanent, seasonal or based on water frequency, since no dynamic information is provided in the attributes and this might be useful in routing calculation.

    Response: Thank you so much for this suggestion. We've introduced a new attribute "Lperm_glcp" to the lake feature, which quantifies the average percentage of permanent water in each lake. This "Lperm_glcp" was calculated using six years of permanent and total lake water areas obtained from the GLCP database (Meyer et al., 2020; doi:10.1038/s41597-020-0517-4), which was derived from the Global Surface Water (GSW) dataset (Pekel et al., 2016). A higher permanent percentage indicates this lake is more stable or permanent, whereas a lower permanent percentage indicates the lake is more seasonal or intermittent. The following changes have been made in the manuscript to accommodate this new attribute:

    1)  We've added the new attribute "Lperm_glcp" to the lakes feature in Table 1.
    2)  Several sentences have been added to describe the new attribute at the end of section 2.1 (Line 205-211).
    3)  To describe the source dataset (i.e., GLCP), a few sentences have been added in section 3.1 (Line 332-335) and in section 3.1.1 (Line 346-354).
    4)  The details of the calculation process have been provided at the end of section 3.2.6 (Line 616-627).

5.  Table 1. Basin_id stands for ID of the lake-network basin this reach belongs to in the feature component of inter-lake reaches, while in Lake-network basins, it stands for Lake-network basin ID. Please use consistent description if they refer to the same attribute, or use different names if they are separate attributes.

Response: Thank you so much for noticing the issue. "Basin_id" in inter-lake reaches and "Basins_id" in lake-network basins are the same attribute. We've updated the description of the "Basin_id" of lake-network basins to "ID of the lake-network basin" in Table 1 to be consistent with other features.

6. While introducing the input data in the 3.1 section, please clarify whether any selection or preprocessing of lake boundaries was conducted. Due to the confusing structure of the article, it is not clear whether the entire HydroLAKES dataset was included.

Response: We appreciate this feedback. We used almost the entire HydroLAKES dataset to generate TopoCat, and we did not perform any preprocessing of the lake boundaries.

At the beginning of the data description section (2.1), we clarified: "*Lake boundaries in TopoCat are geometrically the same as the HydroLAKES lake polygons, except that the former includes new attributes informing lake drainage relations. About 99.95% of the ~1.4 million HydroLAKES polygons were located within or on the MERIT Hydro boundary. These lakes were used to construct TopoCat (Fig. 2a-b).*" (Line 143-145)

In addition, we have updated the structure of this paper to make it more readable. Now the input data is a part of the methodology section, and validation data is a part of the quality assessment and validation section. Please see the details in our response to your comment #2. Furthermore, the second sentence of the dataset development and algorithm section (i.e., section 3.2 of the updated manuscript) has been revised as follows to clearly explain how we utilized the HydroLAKES dataset:

"*In brief, the algorithm started with an iterative process that used MERIT Hydro's flow accumulation layer to identify all possible outlets for each HydroLAKES lake (i.e., pour points to all possible outbound directions from this lake) and each of their associated unit catchments.*" (Line 385-387)

7. Line 415. It would be better if a brief description of how the clusters in Fig. 9 were generated is provided.

Response: A brief description has been added to explain how we produced the clusters of islands in Figure 9 as follows:

"*We carefully grouped these islands in such a way that the area of the rectangular box that covers each group is not larger than the largest extent of the original 61 level-2 basins.*" (Line 412-413)

Figures:

Fig. 4. It would be better if a pie chart as in Fig. 2 is provided.

Response: Thanks for the suggestion. We've added a pie chart to Figure 4. Please also see the response to reviewer#1.